

# The material properties of ice bridges in the Maxwell Elasto-Brittle rheology

Mathieu Plante[1], Bruno Tremblay[1], Martin Losch[2], and Jean-François Lemieux[3]

[1]Department of Atmospheric and Oceanic Sciences, McGill University, Montréal, Québec, Canada
[2]Alfred Wegener Institute for Polar and Marine Research, Bremerhaven, Germany
[3]Recherche en prévision numérique environnementale, Environnement et changement climatique Canada, Dorval, Québec, Canada.

**Correspondence:** Mathieu Plante (mathieu.plante@mail.mcgill.ca)

**Abstract.**

The shape and break-up of landfast ice arches in narrow channels depend on the material properties of the sea-ice. The effect of the material parameters on ice arches in a sea ice model with the Maxwell Elasto-Brittle (MEB) rheology is investigated. The MEB rheology, which includes a damage parameterization, is implemented using the numerical framework of a Viscous-Plastic model. This configuration allows to study their different physics independently of their numerical implementation. Idealized ice bridge simulations show that the elastic part of the model together with the damage parameterization allows the propagation of fractures in space at very short time-scales. The fractures orientation is sensitive to the chosen angle of internal friction, but deviates from theory. It is speculated that these deviations stem from the absence of a flow rule in the rheology. Downwind of a channel, the MEB model easily forms ice arches and sustains an ice bridge. Using a material cohesion in the range of 15-21 kPa is most consistent with the ice bridges commonly observed in the Arctic. Upstream of the channel, the formation of ice arches is complicated by the absence of a relationship between the ice strength and the ice conditions, and by the presence of numerical errors associated with the damage parameterization. Results suggest that the formation of ice arches upwind of a channel is highly dependent on the rheology and calls for more analysis to determine the necessary conditions for their formation.

## 1 Introduction

The term landfast ice designates sea ice that is attached to the coastlines, acting as an immobile and seasonal extension of the land. It starts to form in shallow water in the early stages of the Arctic freeze up (Barry et al., 1979; Reimnitz et al., 1978) and grows throughout the Arctic winter, usually reaching its maximum extent in early spring (Yu et al., 2014). Typically, large landfast ice areas are formed when protected from offshore sea ice dynamics, either by the presence of islands or by the grounding of ice keels on the ocean floor (Reimnitz et al., 1978; Mahoney et al., 2007; Selyuzhenok et al., 2017). Where the





water is too deep for grounding, landfast ice also forms where ice floes are jammed in narrow passages between islands or pieces of grounded ice. In the Canadian Arctic Archipelago (CAA), this type of ice is referred to as land-locked. The resulting ice bridges, also called ice arches for their characteristic arching edges (Fig. 1), can have a profound influence on sea ice

circulation (Melling, 2002; Kwok, 2005) and regional oceanography (Barber and Massom, 2007; Dumont et al., 2010; Shroyer et al., 2015). Most studies about land-locked ice focused on the Nares Strait and Lincoln Sea ice bridges (Kozo 1991; Moore and McNeil 2018), which affect the export of thick multi-year ice into the Baffin Bay (Kwok and Cunningham, 2010; Ryan and Münchow, 2017). Ice arches however are a seasonal feature in several locations of the Canadian Arctic Archipelago (Marko and Thomson 1977; Sodhi 1997; Melling 2002). They are linked to local landfast ice extensions in the peripheral seas (e.g. in

the Kara Sea Divine et al. 2004; Ólason 2016) and likely play a role in the formation of the extensive landfast ice cover of the Laptev Sea (Selyuzhenok et al., 2015).

  Despite decades of observations (Melling, 2002; Kwok, 2005; Moore and McNeil, 2018; Ryan and Münchow, 2017), the formation, persistence and break up of ice arches remain difficult to explain. In particular, ice arches in the Nares Strait (Fig. 1) are largely unpredictable (Melling, 2002; Ryan and Münchow, 2017). A variety of studies suggest that ice arches are influenced

by several factors, such as ice thickness anomalies, enhanced northerly winds, tides or the presence of icebergs (Kwok, 2005; Samelson et al., 2006; Moore and McNeil, 2018). If case studies can attribute ice arch formation or break up anomalies to a combination of these factors, other cases with different behaviour can be found under similar conditions (Moore and McNeil, 2018).

  It is however clear, especially via modeling studies, that the ability of sea ice to form arches relates to the material properties

of sea ice. A number of studies showed that ice arches are produced if the rheology includes sufficient material cohesion (Ip, 1993; Hibler et al., 2006; Dumont et al., 2008). Using the ellipse yield curve of Hibler (1979), this can be done either by decreasing the ellipse ratio (Kubat et al., 2006; Dumont et al., 2008)) and/or by extending the ellipse towards larger isotropic tensile strength (Beatty and Holland, 2010; Ólason, 2016; Lemieux et al., 2016). The range of parameter values that are appropriate for the production of ice bridges varies between different models, suggesting that other model components may

influence the ice arch formation, such as the minimum viscosity (Ólason, 2016), ice grounding (Lemieux et al., 2016) and the presence of tides (Lemieux et al., 2018).

  In recent years, new rheology were proposed to better represent the observed characteristics of ice failure, such as the preferred orientation of the lines of fracture (Wilchinsky and Feltham, 2004; Schreyer et al., 2006), or the brittle behaviour of sea ice at small scales (Girard et al., 2011; Dansereau et al., 2016). In particular, a brittle damage parameterization (Amitrano

et al., 1999) was implemented for sea ice modelling in the neXtSIM model (Rampal et al., 2015). The damage parameterization is based on the notion of progressive damage, originally developed in rock mechanic models to reproduce the non-linear (brittle) behaviour in rock deformation and seismicity (Cowie et al., 1993; Tang, 1997; Amitrano and Helmstetter, 2006). In these models, the material damage associated with micro-cracking is simulated by altering the material properties (e.g. the Young Modulus or the material strength) at the model element scale. If heterogeneity is present in the material, the damage

parameterization simulates the self-organisation of the microcracks in a macroscopic line of fracture, as observed in laboratory experiments. It was first used for large scale sea ice modeling by Girard et al. (2011) as the basis for a new Elasto-Brittle (EB)





rheology, later implemented in the fully Lagrangian dynamic-thermodynamic sea ice model neXtSIM (Rampal et al., 2015). The EB rheology was shown to reproduce the statistical and scaling properties of sea ice deformations (Girard et al., 2011), but was limited by the fact that its linear-elasticity only produces reversible deformations. This problem was addressed in the

Maxwell Elasto-Brittle rheology (Dansereau et al., 2016) by including a viscous term in the constitutive relation, dissipating the elastic stresses into permanent deformations in the manner of a Maxwell visco-elastic material. The MEB rheology has been shown to reproduce the statistical characteristics of sea ice deformations, such as intermittency, fracture localization and fractality (Dansereau et al., 2017), and is now used in the most recent version of neXtSIM (Rampal et al., 2019).

Dansereau et al. (2017) used idealised and realistic Nares Strait ice bridge simulations to evaluate characteristics of the ice

fracture at the geophysical scale produced by the MEB model. The rheology was shown to produce multiple ice arches in Nares Strait. The ice arches in Dansereau et al. (2017) tended to be located downwind of the channels, at the edge of either Smith Sound or Kane Basin (see their Figure 7). Although ice arches downwind of narrow channels are observed in other parts of the CAA (e.g. in the Lancaster Sound, Fig 1a), this location differs from the observed ice arch position in the Nares Strait, upwind of constriction points (e.g. see Fig 1b). This behaviour is also different from the ice arches simulated using the VP or EVP

models, which are also formed upwind of the constriction points (e.g. Dumont et al. 2008; Rasmussen et al. 2010). Since both the MEB and EVP models have a visco-elastic relationship, this different behavior is probably related either to the different fracture parameterization (i.e. Brittle in the MEB, plastic in the EVP) or to the different numerical frameworks (finite element methods in Dansereau et al. 2017, finite difference in Dumont et al. 2008).

In this paper, we investigate the role of the damage parameterization and the material strength parameters in the formation

of ice arches using a MEB rheology that we implemented in an Eulerian finite difference VP model. To our knowledge, it is the first time the MEB rheology is implemented in a finite difference framework, such that the different physics can be assessed independently from the numerical implementation. Using idealised channel simulations, we show that the stress concentration, and thus memory, is dominant in determining the localisation of the ice fractures. We show that a rheology with a dependency between the ice strength and the ice thickness is needed for realistic ridge building. We also show that the simple

stress correction used in the damage parameterization corresponds to a flow rule that is independent of the orientation of the failure surface. The stress correction scheme also amplifies numerical convergence errors by orders of magnitude for large compressive states, introducing numerical artifacts that accumulate in the simulated fields.

This paper is organised as follows. First, we present our implementation of the Maxwell Elasto-Brittle rheology in our Finite Difference framework (section 2). We then present a detailed analysis of the break up of the ice bridge simulated by the MEB

rheology (section 3), along with a sensitivity analysis of the results with respect to the yield and other model parameters. Conclusions are summarized in section 4.





## 2 Maxwell Elasto-Brittle Model

### 2.1 Momentum and continuity equations

The simplified momentum equation for the 2D motion of sea ice used in this model is :

$$\rho_i h \frac{\partial \mathbf{u_i}}{\partial t} = \nabla \cdot \sigma + \tau_a - \tau_w, \tag{1}$$

where $\rho_i$ is the ice density, $h$ is the mean ice thickness, $\mathbf{u}$ ($= u\hat{\mathbf{i}} + v\hat{\mathbf{j}}$) is the ice velocity vector, $\sigma$ is the vertically integrated internal stress tensor, $\tau_{\mathbf{a}}$ ($= \tau_{ax}\hat{\mathbf{i}} + \tau_{ay}\hat{\mathbf{j}}$) is the air-ice surface stress and $\tau_{\mathbf{w}}$ ($= \tau_{wx}\hat{\mathbf{i}} + \tau_{wy}\hat{\mathbf{j}}$) is the ice-ocean stress. The advection of momentum is neglected, being orders of magnitude smaller than the other terms in the momentum balance (Zhang and Hibler, 1997; Hunke and Dukowicz, 1997). The ocean current, and thus the sea surface tilt term, is set to zero and the Coriolis term is ignored, as it is identically zero for landfast ice. These simplifications are reasonable for landfast ice and keep the axial symmetry of the problem. The air-ice stress $\tau_{\mathbf{a}}$ and ice-water surface stress $\tau_{\mathbf{w}}$ terms are defined using standard bulk formula (McPhee, 1979), with the air and ocean turning angles set to zero and assuming that the wind velocity is orders of magnitude larger that of sea ice:

$$\tau_a = \rho_a C_{da} |\mathbf{u}_a| \mathbf{u}_a, \tag{2}$$

$$\tau_w = \rho_w C_{dw} |\mathbf{u}_i| \mathbf{u}_i, \tag{3}$$

where $C_{da}$ and $C_{dw}$ are the air and water drag coefficients, $\mathbf{u}_a$ is the surface wind and $\rho_a$ and $\rho_w$ are the air and water density (see values in Table 2).

The continuity equations used for the temporal evolution of mean ice thickness $h$ (volume per grid cell area) and concentration $A$ are written as:

$$\frac{\partial h}{\partial t} + \nabla \cdot (h\mathbf{u}) = S_h, \tag{4}$$

$$\frac{\partial A}{\partial t} + \nabla \cdot (A\mathbf{u}) = S_A, \tag{5}$$

where $S_h$ and $S_A$ are thermodynamic sink and source terms for ice thickness and compactness respectively. For simplicity, $S_h$ and $S_A$ are set to zero in this study.

### 2.2 Rheology

Following (Dansereau et al., 2016), we consider the ice as a visco-elastic-brittle material, behaving like a spring and dash-pot in series if the stresses are relatively small (Maxwell part), and breaking when larger stresses are present, after which





large deformations are possible. The constitutive equation of the Maxwell Elasto Brittle (MEB) model follows the Maxwell
rheological model :

$$
\frac{\partial \sigma}{\partial t} + \frac{1}{\lambda}\sigma = EC : \dot{\epsilon},
\tag{6}
$$

where $E$ is the Elastic Stiffness defined as the vertically integrated Young Modulus of sea ice, $\lambda$ is the viscous time relaxation
($\lambda = \frac{\eta}{E}$, $\eta$ being the viscosity) and $\dot{\epsilon}$ is the strain rate tensor. The advection of stress is neglected in this study as we focus on
the landfast ice. In the linear elastic regime (when the main balance is between the first term on the lhs and the rhs of Eq. 6),
deformations are relatively small, occur over a small time-scale and are reversible. As such, the elastic strains return to zero if
all loads are removed and the model includes a material memory of the strain history. In the viscous-elastic regime (when all
three terms are important), viscous deformations, or creep, relax the elastic stresses over a longer time-scale if the forcing is
sustained, which results in permanent deformations. The brittle regime occurs when the internal stresses reach a material yield
criterion. In this case, brittle fractures (or cracks) progressively reduce the elastic stiffness and visco-elastic ratio, allowing
for larger visco-elastic deformations with the same amount of stress. This brittle component is thus comparable to the plastic
regime of the standard VP model of Hibler (1979), with deformations that are relatively large, and stresses that are strain rate
independent.

Note that while Eq. 6 is similar in form to the stress-strain relationship of the Elastic Viscous Plastic (EVP) model (Hunke
and Dukowicz, 2002), the elastic component in the EVP model does not represent the true elastic behavior of sea ice. Rather,
the elastic term was introduced to improve the model convergence and computational efficiency of the VP model, allowing for
an explicit numerical scheme and simple parallelization (Hunke and Dukowicz, 1997). In fact, in the new implementation of the
EVP model (Hunke, 2001), elastic waves are faster than the observed elastic waves unless a time-step of several hours or more
is used (Williams et al., 2017). In the MEB model, the elastic component represents the true elastic behavior of sea ice while the
viscous relaxation component is introduced to dissipate the elastic strains into permanent deformations. The use of a viscous
component is consistent with the observation of viscous creep (Tabata, 1955; Weeks and Assur, 1967) and viscous relaxation
(Tucker and Perovich, 1992; Sukhorukov, 1996; Hata and Tremblay, 2015b) in field experiments. The viscous relaxation term
is also analogous to the viscous term in the thermal stress models of Lewis (1993) and Hata and Tremblay (2015a).

### 2.2.1 Linear Elasticity

The elastic component of the MEB rheology implies that the internal stresses are related to the strains (units of $m/m$) rather
than to the strain rates (units of $s^{-1}$). We first write the stress-strain relationship of a 2-D linear elastic solid, that is, ignoring
the viscous and brittle components of the rheology (Rice, 2010; Bouillon and Rampal, 2015)) :

$$
\sigma_{ij} = \frac{E\nu}{1-\nu^2}\delta_{ij}\epsilon_{kk} + \frac{E(1-\nu)}{1-\nu^2}\epsilon_{ij},
\tag{7}
$$

where $E$ is the Elastic Stiffness of sea ice, $\sigma_{ij}$ are the vertically integrated stresses acting in the $j^{th}$ direction on a plane
perpendicular to the $i^{th}$ direction, $\nu$ ( $= 0.33$) is the Poisson ratio and $\delta_{ij}$ is the Kronecker delta. The linear elastic stress





component of Eq. 6 is obtained by taking the time derivative of Eq (7), assuming negligible variations in $E$:

$$\frac{\partial \sigma}{\partial t} = EC : \dot{\epsilon}, \tag{8}$$

where ":" denotes an inner double tensor product and where

$$C = \frac{1}{1-\nu^2} \begin{pmatrix} 1 & \nu & 0 \\ \nu & 1 & 0 \\ 0 & 0 & 1-\nu \end{pmatrix} = \begin{pmatrix} C_1 & C_2 & 0 \\ C_2 & C_1 & 0 \\ 0 & 0 & C_3 \end{pmatrix} \tag{9}$$

and

$$\dot{\epsilon}_{ij} = \frac{1}{2}\left[\frac{\partial u_i}{\partial x_j} + \frac{\partial u_j}{\partial x_i}\right] \quad or \quad \begin{pmatrix} \dot{\epsilon}_{11} \\ \dot{\epsilon}_{22} \\ \dot{\epsilon}_{12} \end{pmatrix} = \begin{pmatrix} \frac{\partial u}{\partial x} \\ \frac{\partial v}{\partial y} \\ \frac{1}{2}\left(\frac{\partial u}{\partial y} + \frac{\partial v}{\partial x}\right). \end{pmatrix} \tag{10}$$

The components of the tensor $C$ are derived using the plane stress approximation, (i.e. following the original assumption that the vertical stress components are negligible, see for instance Rice 2010). This form is also used in Bouillon and Rampal (2015); Rampal et al. (2015), but differs from Dansereau et al. (2015) who used the components derived from the plane strain approximation, in which case the vertical components of strain are zero but the vertical normal stress component ($\sigma_{33}$) is non-

zero. This should be avoided, as it implies that the vertical stress is of significant order (Rice, 2010) and determined by the horizontal normal stress: $\sigma_{33} = \nu(\sigma_{11} + \sigma_{22})$.

### 2.2.2   Brittle fracture parameterization

In this model, the critical stress at which the ice fails is determined by the Mohr-Coulomb failure criterion. This yield criterion is based on laboratory experiments (Schulson et al., 2006) and was found to agree with field observations (Weiss et al., 2007).

A criterion is further used on the second principal stress ($\sigma_2 = \sigma_I - \sigma II$) to limit the compression (see Fig. 2). The yield function can be written in terms of the stress invariant $\sigma_I$ and $\sigma_{II}$, as:

$$F(\sigma) = \begin{cases} \sigma_{II} + \mu\sigma_I - c < 0 & \text{Mohr Coulomb} \\ \sigma_I - \sigma_{II} > \sigma_c & \text{Compression cut-off} \end{cases} \tag{11}$$

where $\sigma_I$ is the isotropic normal stress (compression defined as negative), $\sigma_{II}$ is the maximum shear stress, $c$ is the cohesion, $\mu = \sin\phi$ is the coefficient of internal friction of ice, $\phi$ is the angle of internal friction and $\sigma_c$ is the compressive strength.

The cohesion, angle of internal friction and compressive strength are material properties derived from in-situ observations (see Table 1 for values and references) and laboratory experiments (Timco and Weeks, 2010). The model values used in our control run are listed in Table 2 along with a summary of the value used in other studies in Table 3.

    The fracture of ice is parameterized as a local decrease in elastic stiffness and consequently as a local increase in the magnitude of the elastic deformation associated with a given stress state. The local increase in deformations in the material

results in the concentration of internal stresses in adjacent grid cells, leading to the propagation of the brittle fracture in space.





The elastic stiffness is written in function of a damage parameter $d$ representing the weakening of the ice upon fracturing ($0 < d \leq 1$, Bouillon and Rampal 2015), with a dependency on the ice thickness and sea ice concentration inspired by the ice strength parameterization of Hibler (1979) :

$$E = Yhe^{-C(1-A)}(1-d),\tag{12}$$

where Y (= 1 GPa) is the Young Modulus of undeformed sea ice. This value is smaller than that used in the neXtSIM model (10 GPA, Bouillon and Rampal 2015) and similar to that of the MEB model (0.8 GPa, Dansereau et al. 2016). The damage parameter $d$ has a value of 0 for undamaged sea ice and 1 for fully damaged ice, and represents the amount of fractures and material degradation present in the ice. Note that the yield parameters ($c$, $\mu$, $\sigma_c$) in the MEB model are not a function of the damage parameter (nor of the ice thickness or concentration). The decrease in elastic stiffness when the ice fails yields larger
deformations (and potentially changes in h and A) for the same state of stress, but does not change the critical stress (i.e., the ice strength). This is in contrast with a strain-weakening or strain-hardening material, a property that has been observed in sea ice (Richter-Menge et al., 2002). The lack of strain hardening in the MEB model leads to non-physical results in convergence with the absence of ridge propagation in the direction parallel to the second principal strain (maximum axial compressive strain). This will be discussed in section 3.1.4.

Following Rampal et al. (2015), the introduction of damage upon failure is proportional to the local stress in excess of the yield criterion. A damage factor $\Psi$ ($0 < \Psi < 1$) is used to keep the stress on the yield curve. It is defined as (see appendix A for the derivation $\Psi$) :

$$\sigma_f = \Psi\sigma' \qquad \text{with} \qquad \Psi = \min\left(1, \frac{c}{\sigma'_{II} + \mu\sigma'_I}\right),\tag{13}$$

where $\sigma_f$ is the corrected stress lying on the yield curve and $\sigma'$ is the prior stress state that exceeds the yield criterion. In the
following, prime quantities are used to indicate stress terms that are not corrected by the damage factor $\Psi$. Note that the stress components are all scaled by the same damage factor, following a line from the uncorrected stress state to the origin (see Fig. 2). This correction does not corresponds to a flow rule: only the magnitude of the excess stress is used to increase the damage parameter. This determines the magnitude of the strain associated to a stress state (and indirectly, the strain rate tensor), but not its orientation. This is different from the normal flow rule of Hibler 1979 (see for instance Bouchat and Tremblay 2017), in
which the yield curve surface determines the relative importance of shear and divergence in the deformations.

  After each fracture event, the damage parameter is updated such that $(1 - d_f) = \Psi(1 - d')$. That is, the damage factors are accumulated in the damage parameter during the simulation, such that the damage gradually increases with each fracture. As the amount of stress overshoot is time-step dependent, the temporal evolution of the damage parameter is parameterized as a simple relaxation with a damage time scale $T_d$ (Dansereau et al., 2016):

$$\frac{\partial d}{\partial t} = \frac{(1 - \Psi)(1 - d)}{T_d},\tag{14}$$

where $T_d$ is set to the advective time scale associated with the propagation of elastic waves in undamaged ice ($T_d = \Delta x/c_e$, where $\Delta x$ is the spatial resolution of the model and $c_e$ the elastic wave speed). Note that the relaxation time scale in Eq.



(14) ($T_d/(\Psi - 1)$) is time-step dependent via its dependency on the damage factor $\Psi$: a larger time step yields larger stress increments and larger excess stresses at each time-level, decreasing the time scale for the damage relaxation. The sensitivity of

the damage parameterization on the model time step lead Dansereau et al. (2016) to argue that the model time step should be set to exactly $T_d$, otherwise the damage could travel faster than the elastic waves. We argue that while this point is true when using a fixed damage reduction parameter (as in Amitrano et al. 1999; Girard et al. 2011), the use of a damage factor $\psi$ relates the damage parameter to the rate of changes in the stress state, which are associated with the propagation of elastic waves. The propagation of damage in space is thus bounded by the elastic wave speed, and a smaller time-step should be favored if

possible in order to resolve the elastic waves.

Note that no damage healing process was included in this study as we focus on the break up of ice bridges at small time scales. For the same reason, the advection of damage is neglected.

### 2.2.3 Maxwell viscosity

The viscosity $\eta$ used to define the viscous time relaxation ($\lambda = \frac{\eta}{E}$) is parameterized to change faster with the damage parameter

than the elastic stiffness (Dansereau et al., 2016) :

$$\eta = \eta_0 (1 - d)^\alpha \tag{15}$$

with $\eta_0$ being the viscosity of undeformed sea ice and $\alpha$ an integer set to 4 that determines the smoothness of the transition from linear elastic behavior to viscous behavior. This definition ensures that the viscous term is negligible in undamaged ice, but important in heavily damaged ice (see Eq. 6, where $\eta$ appears in the denominator). Note that as the damage increases

asymptotically to 1, the singularity in Eq. 15 (if $d = 1$) never occurs. The relaxation time $\lambda$ in Eq. 6 can then be written using Eq. 12 and 15 :

$$\lambda = \frac{\eta}{E} = \frac{\lambda_0 (1 - d)^{\alpha - 1}}{h e^{-C(1-A)}}, \tag{16}$$

where $\lambda_0$ ($= \eta_0/Y = 10^5$) is a parameter that corresponds to the viscous relaxation time scale in undamaged sea ice with 1m mean thickness. The value of about one day falls into the range of observations (see Table 1). Note that if $\lambda_0$ is sufficiently

high, the MEB rheology reduces to the Elasto-Brittle rheology (Bouillon and Rampal, 2015; Rampal et al., 2015). The set dependency of $\lambda$ on the mean ice thickness and concentration ensures that both the total stress tends toward zero for low mean thickness or concentration (i.e. in free drift), while a continuous ($A \sim 1$, $h > 0$) but heavily damaged ice cover behaves as a viscous material.

### 2.3 Numerical approaches

In this section, we discuss the numerical implementation of the MEB rheology exploiting the code framework of a standard VP model. In this framework, the stress components in the momentum equation do not appear explicitly; instead they are written in terms of viscous coefficients and strain-rates. This is done in our implementation of the MEB model by treating the stress memory as an additional forcing. The damage parameterization is therefore the only new module that needs to be implemented.



### 2.3.1 Time discretization

The model equations are discretized in time using a semi-implicit backward Euler scheme. The uncorrected stress at time level $n$ can then be written using Eq. 6, as:

$$\sigma'^n = \frac{1}{1+\Delta t/\lambda^n}\left[E^n \Delta t C : \dot{\epsilon}^n + \sigma^{n-1}\right] = \xi^n C : \dot{\epsilon}^n + \gamma^n \sigma^{n-1}, \tag{17}$$

where $n-1$ is the previous time level and where:

$$\xi^n = \gamma^n E^n \Delta t \qquad ; \qquad \gamma^n = (1+\Delta t/\lambda^n)^{-1}. \tag{18}$$

Note that $\sigma^n$ is a function of $\sigma^{n-1}$, which we refer to as the stress memory. Equation 17 is then substituted in the stress divergence term of Eq. 1, so that the x and y components of the momentum equation can be expanded as :

$$\rho_i h^n \frac{u^n - u^{n-1}}{\Delta t} = \frac{\partial}{\partial x}\left(\xi^n C_1 \epsilon_{xx}^n\right) + \frac{\partial}{\partial x}\left(\xi^n C_2 \epsilon_{yy}^n\right) + \frac{\partial}{\partial y}\left(\xi^n C_3 \epsilon_{xy}^n\right) + \tau_x^n, \tag{19}$$

$$\rho_i h^n \frac{v^n - v^{n-1}}{\Delta t} = \frac{\partial}{\partial y}\left(\xi^n C_1 \epsilon_{yy}^n\right) + \frac{\partial}{\partial y}\left(\xi^n C_2 \epsilon_{xx}^n\right) + \frac{\partial}{\partial x}\left(\xi^n C_3 \epsilon_{xy}^n\right) + \tau_y^n, \tag{20}$$

where $C_1$, $C_2$, and $C_3$ are the components of the tensor $C$ (Eq. 9) and where the stress memory terms have been included in the forcing, that is :

$$\tau_x^n = \frac{\partial\left(\gamma^n \sigma_{xx}^{n-1}\right)}{\partial x} + \frac{\partial\left(\gamma^n \sigma_{xy}^{n-1}\right)}{\partial y} + \tau_{ax}^n + \tau_{wx}^n, \tag{21}$$

$$\tau_y^n = \frac{\partial\left(\gamma^n \sigma_{yy}^{n-1}\right)}{\partial y} + \frac{\partial\left(\gamma^n \sigma_{xy}^{n-1}\right)}{\partial x} + \tau_{ay}^n + \tau_{wy}^n. \tag{22}$$

The MEB rheology equations can then be implemented in a VP model by setting the VP bulk and shear viscosity to $\zeta_{VP} = \xi\frac{C_1+C2}{2}$ and $\eta_{VP} = \xi C_3$ respectively, setting the pressure term to $P = 0$ and adding the stress memory terms.

The variable $h^n$, $A^n$ and $d^n$ (and thus $E^n$ and $\lambda^n$) in Eq. 17 to 20 are discretized explicitely, as:

$$h^n = h^{n-1} + \nabla \cdot (\mathbf{v}^n h^{n-1} \Delta t), \tag{23}$$

$$A^n = A^{n-1} + \nabla \cdot (\mathbf{v}^n A^{n-1} \Delta t), \tag{24}$$

$$d^n = d^{n-1} + \frac{d^{n-1}\Delta t}{T_d}(\Psi^n - 1), \tag{25}$$





$$E^n = E_0 h^n d^n e^{-c(1-A^n)},\tag{26}$$


$$\lambda^{n,k} = \frac{\lambda_0 (d^n)^{\alpha-1}}{h^n e^{-C(1-A^n)}},\tag{27}$$

where the damage factor is computed from Eq. 13 and Eq. 17.

### 2.3.2 Space discretization

The model equations are discretized in space using a centered finite different scheme on an Arakawa C-grid. In this grid, the
diagonal terms of the $\sigma$ and $\dot{\epsilon}$ tensors are naturally computed at the cell centers and the off-diagonal terms at the grid nodes
(see Fig. 3). The x-component of the momentum equation are written as :

$$\rho_i h_{i,j}^{n-1} \frac{u_{i,j}^n - u_{i,j}^{n-1}}{\Delta t} = C_1 \frac{\left(\xi^{n-1}\epsilon_{xx}^n\right)_{i,j} - \left(\xi^{n-1}\epsilon_{xx}^n\right)_{i-1,j}}{\Delta x} + C_2 \frac{\left(\xi^{n-1}\epsilon_{yy}^n\right)_{i,j} - \left(\xi^{n-1}\epsilon_{yy}^n\right)_{i-1,j}}{\Delta x}$$
$$+ C_3 \frac{\left(\xi_z^{n-1}\epsilon_{xy}^n\right)_{i,j+1} - \left(\xi_z^{n-1}\epsilon_{xy}^n\right)_{i,j}}{\Delta y} + \tau_{x\,i,j}^n \tag{28}$$

where :

$$(\dot{\epsilon}_{xx}^n)_{i,j} = \frac{u_{i+1,j}^n - u_{i,j}^n}{\Delta x},\tag{29}$$


$$(\dot{\epsilon}_{yy}^n)_{i,j} = \frac{v_{i,j+1}^n - v_{i,j}^n}{\Delta y},\tag{30}$$

$$(\dot{\epsilon}_{xy}^n)_{i,j} = \frac{u_{i,j}^n - u_{i,j-1}^n}{2\Delta y} + \frac{v_{i,j}^n - v_{i-1,j}^n}{2\Delta x},\tag{31}$$

$$\tau_{x\,i,j}^n = \frac{\left(\gamma^{n-1}\sigma_{xx}^{n-1}\right)_{i,j} - \left(\gamma^{n-1}\sigma_{xx}^{n-1}\right)_{i-1,j}}{\Delta x} + \frac{\left(\gamma_z^{n-1}\sigma_{xy}^{n-1}\right)_{i,j+1} - \left(\gamma_z^{n-1}\sigma_{xy}^{n-1}\right)_{i,j}}{\Delta y} + \tau_{ax\,i,j}^n + \tau_{wx\,i,j}^n.\tag{32}$$

The shear terms in Eq. 28 and 32 ($\dot{\epsilon}_{xy}$, $\xi_z$ and $\gamma_z$) are thus defined at the lower-left grid node rather than at the grid center.
This staggering of the stress components is unavoidable when using the C-grid, and requires node approximations for the
scalar values $h$, $A$ and $d$ (Losch et al., 2010). This is treated on our Cartesian grid with square cells by approximating the scalar
prognostic variables at the nodes ($h_z$, $A_z$ and $d_z$) as a simple average of the neighbouring cell centres, i.e. :

$$h_z = \bar{h}_{i,j} = \frac{h_{i,j} + h_{i-1,j} + h_{i,j-1} + h_{i-1,j-1}}{4},\tag{33}$$

and similarly for $A_z$ and $d_z$. The stress-strain coefficients $\xi_z$ and $\gamma_z$ are then computed using ($h_z$, $A_z$ and $d_z$) in Eq. 12, 16 and
18.

The shear stress at the cell centre must also be approximated when computing the stress invariants in the stress correction scheme (Eq. 13). Averaging the shear stress components (as in Eq. 33 for the scalars) cause a checker board instability to appear in the solution, because of the staggered shear stress corrections and memories. To avoid this, we approximate the shear stress at the cell center using a different shear stress memory term, defined at the grid center and only used in the damage parameterization, and only average the shear stress increments. That is:

$$\sigma'^{n}_{xy\,i,j}|_C = \overline{\left(\xi^n_z \dot{\epsilon}^n_{xy}\right)}_{i,j} + \gamma^{n-1}\sigma^{n-1}_{xy\,i,j}|_C,\tag{34}$$

where $\sigma'^{n}_{xy\,i,j}|_C$ is the uncorrected shear stress at the grid center and $\sigma^{n-1}_{xy\,i,j}|_C$ is the corrected shear stress at the grid center from the previous time step. This approximation is only used to calculate the damage factor $\psi$.

Note that the averages in Eq. 33 and 34 cause a smoothing of the variables used to define the shear stress state. This may yield significant differences with the previous implementation of the damage parameterization, which was developed for models using a Lagrangian (Rampal et al., 2015) or Finite Element Method (Dansereau et al., 2016) schemes, but would allow insightful comparisons with VP and EVP model simulations using similar discretization schemes.

### 2.3.3 Numerical solution

The discretized equations described above are solved simultaneously using an IMplicit-EXplicit (IMEX) approach (Lemieux et al., 2014). This process is based on a Picard solver (Lemieux et al., 2008) which involves an Outer Loop (OL) iteration. At each OL iteration $k$, the non-linear system of equations (momentum) is linearized and solved using a preconditioned Flexible General Minimum RESidual method (FGMRES). The latest iterate $u^k$ is used to solve explicitly the damage and continuity equations until the L2norm of the solution residual falls below a set tolerance of $\epsilon_{res} = 10^{-10}$ N/m$^2$. The uncorrected stresses $\sigma'^n$ is then scaled by the damage factor $\Psi^n$ and stored as the stress memory $\sigma^n$ for the following time step.

To summarize, the IMEX stepping scheme can be written as :

1. Start time level n with initial iterate $\mathbf{u_0}$

do $k = 1, k_{max}$

    2. Linearize the momentum equation using $\mathbf{u}^{n,k-1}$, $h^{n,k-1}$, $A^{n,k-1}$ and $d^{n,k-1}$

    3. Calculate $\mathbf{u}^{k,n}$ by solving Eq. 19 and 20 with GMRES

    4. Calculate $\Psi^{n,k} = f(\sigma'^{n,k})$

    5. Calculate $h^{k,n} = f(h^{n,k-1}, \mathbf{u}^{n,k})$, $A^{n,k} = f(A^{n,k-1}, \mathbf{u}^{n,k})$, $d^{n,k} = f(d^{n,k-1}, \mathbf{u}^{n,k}, \Psi^{n,k})$

    6. Calculate $E^{k,n} = f(d^{n,k}, h^{n,k}, A^{n,k})$, $\lambda^{n,k} = f(d^{n,k}, h^{n,k}, A^{n,k})$

    7. If the Picard solver converged to a residual $< \epsilon_{res}$, stop.

enddo

8. Update the stress memory $\sigma^n = \Psi^n \sigma'^n$



## 3  Results

In the following, we present a series of ideal simulations to document the formation and break-up of ice arches with the
MEB rheology, and their sensitivity to the choice of mechanical strength parameters. Results from these simulations and
observations are used to constrain the material parameters used in sea ice models. Here, we define an ice arch as the location of
the discontinuity in the sea ice velocity–and later in the ice thickness and concentration fields–and the ice bridge as the landfast
ice upwind of the ice arch.

Our model domain is 800 x 200 km with a spatial resolution of 2 km (Fig. 4). The boundary conditions are periodic on the
left and right, closed on the top and open on the bottom. Two islands, separated by a narrow channel 200 km long and 60 km
wide, are located 300 km away from the top and bottom boundaries. The initial conditions for sea ice are zero ice velocity,
uniform 1m ice thickness, 100 % concentration and zero damage. A southward wind forcing is imposed on the ice surface,
and ramped up from 0 m/s to 20 m/s in a 10h period, well below the adjustment time scale associated with elastic waves. The
solution can therefore be considered as steady state at all time, which allows us to determine the critical wind forcing associated
with a fracture event.

### 3.1  Control run

The break up of landfast ice in our simulation proceeds through a series of fracture events that are highly localized in time (see
Fig. 5) and space (see Fig. 6 and 12), separated by periods of elastic stress build up (low brittle failure activity). Two major
fracture events are seen in the simulation (stage B and D in Fig. 5). The first corresponds to the failure of ice in tension with
the development of an ice arch on the downwind side of the channel (Fig. 6). The second event corresponds to the collapse of
the landfast ice bridge with the break up of ice within and upwind of the channel (Fig. 12). The three remaining periods during
which few new brittle fractures occur correspond to an elastic landfast ice regime (stage A), a stable downwind ice arch regime
(stage C), and a drift ice regime when ice flows within, downstream and upstream of the channel (stage E).

#### 3.1.1  Elastic regime: stage A

In the first stage of the simulation, elastic stress builds up but remains inside the yield curve in the entire domain such that
there is no brittle failure activity (Fig. 5, stage A). The sea ice in the elastic regime behaves as an elastic plate and deformations
are linearly related to the internal stresses. The elastic stresses are determined by the orientation of the wind with respect
to the coastlines: there are large tensile stresses on the downwind coastlines, compressive stresses on the upwind coastlines
and shear stresses on the four corners of the channel (Fig. 8). At the vertical line of symmetry (away from channel openings,
Fig. 8, dashed blue line), the simulated stress field is in good agreement with the analytical solutions from a 1D version of
the momentum equation, giving us confidence in the numerical implementation of the model (see Appendix B and Figure
9). Upstream and downstream of the channel, both stress invariants are important, reaching a maximum in magnitude at the
channel corners and decreasing to a local minimum at the center of the channel. In this configuration, the second principal





stress alignment (Fig. 8c) is along the x-direction downwind of the coastlines (where the ice is in uniaxial tension), and along
the y-direction upwind of the coastlines (where the ice is in uniaxial compression). In the downwind end of channel, the second
principal stress alignment takes an arching shape, transitioning to a vertical alignment towards the upwind channel entrance.

### 3.1.2 Downstream ice arch: stage B

The formation of the downstream ice arch (Fig. 6) is initiated at a wind forcing of $\sim 3.5$ m/s. The initial fractures are located
at the downwind corners of the channel where the stress state reaches the critical shear strength for positive (tensile) normal
stresses (Fig. 10). The fractures then propagate from these locations and form an arch (Fig. 6, point 3). The progression of the
fracture into an ice arch is helped by the concentration of stresses at the channel corners and around the subsequent damage.
That is, the damage permanently decreases the elastic stiffness, which leads to locally larger elastic deformations and increases
the load in the surrounding areas. This results in the propagation of the fractures in space through regions where the internal
stress state was originally sub-critical. This process occurs on very short time scales (within minutes), and preconditions the
formation of an arching flaw polynya over longer time scales (Fig. 6b).

To first order, the arching progression of the fracture from the channel corners follows the second principal stress direction
(i.e. a failure in uni-axial tension on the plane perpendicular to the maximum tensile stress, see Fig. 8c). This differs from the
expected angle of fracture in a coulombic material, at $\theta = \pm(\pi/4 - \phi/2)$ from the second principal stress orientation (Ringeisen
et al., 2019). This deviation results from the absence of a flow rule in the MEB model. That is, only the Elastic stiffness is
changed by the damage parameter to scale back the uncorrected stress to the critical state. The strain rate tensor associated
with the fracture is hence determined by the change in stress state at the end of the non-linear solution, rather then by the yield
curve surface (see Fig. 10). The flow rule discrepancy is discussed in more details in section 3.2.2.

### 3.1.3 Stable ice arch regime: Stage C

A second period of low brittle fracture activity follows the formation of the ice arch (period C in Fig. 5). In this stage, the
ice downwind of the ice arch is detached from the land boundaries and starts to drift. Upstream of the ice arch, the elastic
stresses, except for their increase in magnitude due to higher wind forcing, show little changes from stage A. The non-zero
brittle fracture activity in this stage is due to the increased damage in regions of already damaged ice; since the local stress state
lies on the yield curve, the increasing wind forcing constantly increases the stress states beyond the yield criterion, leading to
further damage. Note that unless the yield parameters depend explicitly on the damage, tensile fracturing does not reduce the
critical stress, in contrast to real ice fractures. As such, large tensile and shear stresses persist along and north of the ice arch
after the ice arch is formed (Fig. 8b). The formation of a stress-free surface could be created by defining the cohesion as a
function of the ice thickness and/or damage.



### 3.1.4 Ice bridge collapse: stage D

The second break-up event (Stage D) corresponds to the fracture of ice upwind of the channel and the collapse of the ice bridge
between and upstream of the islands (Fig. 12). This fracture starts at a wind speed of 5.67 m/s on the upwind corners of the
islands where the internal stress reaches the critical shear strength for negative (compressive) normal stresses (green point in
Fig. 10). The propagation of damage from these points is composed of two separate fractures. First, a shear fracture progresses
downwind along the channel walls (Fig. 12, point 5). This results in the decohesion of the landfast ice in the channel from the
channel walls, increasing the load on the ice arch and in the landfast ice north of the channel. Second, a shear fracture propagates
upwind from the channel corners at an angle $61.8°$ from the coastline (Fig. 12, point 6). This orientation corresponds to an
angle $\theta = 28.2°$ from the second principal stress orientation (Fig. 8c). Again, this angle deviates from the theoretical fracture
orientation in a granular material with $\phi = 45°$, at $22.5°$ from the second principal stress orientation (Ringeisen et al., 2019).
As for the downwind ice arch, the lines of fractures are formed at short time scales and precondition the location of ridging on
the advection time scale (Fig. 12b).

### 3.1.5 Drift and ridge building: stage E

The last stage of the simulation (stage E) corresponds to a regime where most of the ice in the domain is drifting. As in Stage
C, the non-zero brittle fracture activity corresponds to further damage being produced in the already damaged ice. In this stage,
landfast ice only remains in two wedges of undeformed ice upwind from the islands (see Fig. 12b). The remaining continuous
areas of undamaged ice drift downward into the funnel as a solid body with uniform velocity, with ridges building at the fracture
lines. Note that the ridge building is highly localised, with no further stress build up elsewhere in the domain. This prevents
the formation of an ice arch upwind of the channel seen in observations (e.g. in the Lincoln Sea). This unrealistic behaviour
of the model is another consequence of the use of constant strength parameters and yield criterion: instead of increasing the
pressure with increasing ice thickness during ridging, the stress field is in a steady state set by the constant critical stress
along the deformation lines (see Fig 13). The increasing wind forcing is then only balanced by the inertial term and water
drag term, resulting virtually in a free drift mode where the ice velocity (and thus the rate of deformation at the ridging lines)
increases with the wind forcing. This causes the ice thickness to increase indefinitely at the ridging location. Rampal et al.
(2015) mitigated this model artifact by the inclusion of a pressure term in the momentum equation. A physical solution to this
problem is to add a dependency between the yield parameters ($c$, $\sigma_c$) and the ice thickness. This would allow compressive
stress to build up along the sliding line and eventually jam the ice. An ice jam would likely result in the formation of an ice
arch upstream of the channel, as suggested by the arching orientation of the second principal stress component in the funnel
(Fig. 13c). This is left as future work.

Note that the damage field at the end of the simulations is highly sensitive to the solution residual tolerance $\epsilon_{res}$. With time,
unless a very low $\epsilon_{res}$ is used, the damage fields are no longer horizontally symmetrical about the center of the channel (Fig.
14). This indicates the presence of artifacts in the model, although the solutions always converged to the set precision. An
error propagation analysis shows that these asymmetries are produced by the computation of the damage factor $\psi$ (Eq. 13).





Assuming that the model is iterated to convergence such that the uncorrected stress state has a relative error of $\epsilon$, the error on the corrected stress is (see derivation in Appendix C):

$$\epsilon_M = \epsilon\sqrt{1+R}, \tag{35}$$

where

$$R = \frac{\sigma_{II}'^2 + \mu^2 \sigma_I'^2}{(\sigma_{II}' + \mu\sigma_I')^2}. \tag{36}$$

If $\sigma_I' > 0$ (tensile stress state), $0 < R < 1$ (triangle inequality) and the error of the memory components ($\epsilon_M$) is of the same order as that of the uncorrected stress state ($\epsilon \leq \epsilon_M \leq \sqrt{2}\epsilon$). If $\sigma_I' < 0$ (compressive stress state), we have $R \geq 1$, and the error on the stress memory can become orders of magnitude larger than that of the uncorrected stress state, and the model accuracy and convergence properties are greatly reduced. These errors are stored in the memory terms, and accumulate at each fracture event. Note that as the elastic stress memory is dissipated over the viscous relaxation time scale, this issue can be improved by decreasing the viscous coefficients $\eta_0$. Another solution would be to use a non-linear yield curve which converges to the Tresca criterion ($\sigma_{II} = $ const) for large compressive stresses (e.g. the yield criterion of Schreyer et al. 2006). We however argue that this issue in the damage parameterization should be treated by bringing the stress back onto the yield curve along a different path (e.g. following a line perpendicular to the curve). A different stress correction path would furthermore allow the application of a flow rule based on granular physics. This would require a damage tensor that would scale the components of the stress tensor independently, as commonly used in continuum damage mechanics. Implementing a damage tensor is left for future work.

## 3.2 Sensitivity to mechanical strength parameters

The Mohr-Coulomb yield criterion defines the shear strength of sea ice as a linear function of the normal stress on the fracture plane. In stress invariant coordinates ($\sigma_I, \sigma_{II}$), this can be written in terms of two material parameters: the cohesion $c$ and the coefficient of internal friction $\mu = \sin\phi$ (Fig. 2). The isotropic tensile strength (i.e. the tip of the yield curve) is then a linear function of the two ($\sigma_t = c/\mu$). In this section, we investigate the influence of these material parameters and of a compression strength criterion on the simulated ice bridge. We place a particular focus on the propagation of the ice fractures in space both upwind and downwind of the channel.

### 3.2.1 Cohesion

Changing the cohesion $c$ with a fixed internal angle of friction $\phi$ moves the entire yield curve along the first stress invariant ($\sigma_I$) axis. For example, a higher cohesion increases the isotropic tensile strength $\sigma_t = c/\sin\phi$ and also increases the shear strength uniformly for all normal stress conditions. In the ice bridge simulations, the choice of cohesion influences the critical wind forcing associated with the different stages of the simulations but does not change the series of events described in section 3 or the orientation of the ice fractures. This is in agreement with results from Dansereau et al. (2017).





The critical wind forcing associated with the ice bridge break up can be related to the cohesion using the 1D steady state momentum equation (see Appendix B for details). Assuming an infinite channel running in the y-direction, the shear stress along the channel walls ($\sigma_{xy}$) is given by:

$$|\sigma_{xy}| = \sigma_{II} = \frac{\tau_a W}{2}, \tag{37}$$

where $W$ is the channel width (see Fig. 4). Using the yield criterion (Eq. 11) with $\sigma_I = 0$ (i.e. $\sigma_{II} = c$), the maximum sustainable wind forcing $\tau_{ac}$ can be related to the cohesion as:

$$\tau_{ac} = \frac{2c}{W}. \tag{38}$$

In the ice bridge simulations, the critical wind forcing for the break up of the ice bridge (stage D) follows the simple 1D model, although with lower wind forcing values (Fig. 15). This is expected considering the stress concentration occurring at
the channel corners and the contribution of the ice upwind and downwind of the non-infinite channel, which pushes and pull on the ice in the channel such that a smaller critical wind forcing is required to break the ice.

Given that ice bridges and arches with a width of $\sim 60$ km are frequent in the CAA (e.g. Nares Strait, Lancaster Sound, or Prince Regent Inlet), and that the wind speed regularly exceeds 10 m/s ($\tau_a > 0.15$ N/m$^2$), this suggests a lower bound on the cohesion of sea ice of at least 15 kN/m (see green curve in Fig. 15). Similarly, the fact that the ice bridges are rarely larger
than 100 km (some are seen intermittently in the Kara Sea, Divine et al. 2004) suggests that the cohesion of sea ice should be smaller than 21 kN/m (see red curve in Fig. 15). These values are lower than estimates from ice stress buoys measurements (40kPa, Weiss et al., 2007) which includes many driving forces neglected in our simulations, such as thermal stresses (Hata and Tremblay, 2015b) and, to a lesser extent, tides. Our values are similar to previous large scale estimates based on wind forcing alone (Tremblay and Hakakian, 2006) and to values used in the neXtSIM model (see Table 3).

### 3.2.2  Angle of internal friction

The angle of internal friction $\phi$, analogous to the static friction between two solids, determines the constant of proportionality ($\mu = \sin\phi$) between the shear strength and the normal stress (see Eq. 11 and Fig. 2). In the following, we vary the angle of internal friction while keeping the cohesion constant. This ensures that the shear strength of ice without confinement (at $\sigma_I = 0$) is the same in the different simulations, so that the critical wind forcing associated with the ice bridge break up remains of
the same order of magnitude. The variations in the angle of internal friction changes the shear strength of ice under tensile and compressive stresses in opposite ways: when increasing the angle of internal friction, the shear strength of ice in tension is reduced while that of ice in compression is increased (and vice versa). This affects the magnitude of the wind forcing associated with downwind and upwind ice fractures, without affecting that of the fractures along the channel walls (start of stage D). That is, with a larger $\phi$, the downwind ice arch (stage B) forms for weaker winds but stronger wind forcing is required
for the development of the upstream lines of fracture and the ice bridge collapse. This shows that while sufficient cohesion is necessary for the stability of the landfast ice in the channel, the collapse of the ice bridge depends on the fracture of ice upstream of the channel, which is sensitive to the angle of internal friction.





In theory, the angle of internal friction governs the intersection angle between lines of fracture (Marko and Thomson, 1977; Pritchard, 1988; Wang, 2007; Ringeisen et al., 2019). That is, the orientation of the failure surface is determined by the

point at which the Mohr circle reaches the yield criterion in the Mohr circle space. This point of failure is aligned at angle $\theta = \pm(\pi/4 - \phi/2)$ from the first principal stress direction (Ringeisen et al., 2019). In the MEB model, the angle of fracture does not follow the theory (see Fig .16b). We speculate that the deviations are related to the absence of a flow rule linking the deformations to the yield curve and the angle of internal friction. Note however that the tendency of the fracture orientation remains consistent with the theory: decreasing the angle of internal friction increases the downwind ice arch curvature and the

angle of the fractures (from the positive x-direction) upwind of the islands. This suggests that despite the absence of a flow rule, the angle of internal friction determines the direction of crack propagation indirectly by changing the material strength.

Note that for angles of internal friction $> 60°$, the upwind lines of fracture propagate away from the boundaries, and a second ice arch forms upwind of the channel (not shown). This angle of internal friction is un-realistically large: previous estimates of the coefficient of friction from ice in situ observations rather suggest values of $30 - 45°$ (see Table 1). This difference stems

from the fact that in the Arctic, the ice arches that are commonly observed upwind of a channel are formed when granular floes jam when forced into a constricting channel in which the ice is not landfast. In our experiments, we rather simulate the propagation of ice fractures through the landfast ice upwind of a channel. It should be possible, however, to form these arches during the drift ice regime after the collapse of the ice bridges. In our simulations, two issues impede this process: the lack of compressive stress build up during ridge building and the flow rule (i.e. the path of the stress correction) which favors ridge

building over sliding along the landfast edges, limiting the increase in compression directly upwind of the channel.

### 3.2.3    Tensile strength

The yield curve modifications discussed above (varying $c$ and $\phi$) also change the tensile strength (both uniaxial and isotropic) of ice. The tensile strength determines the magnitude of the critical wind forcing necessary for the formation of the downwind ice arch (stage B). Downwind from the islands, the tensile stresses can be approximated using the 1D version of the momentum

equation as a function of the fetch distance $F_{down}$ (see Fig. 4) between the downwind coast of the islands and the bottom boundary of the domain (derivation in Appendix B):

$$\sigma_{yy} = \tau_a F_{down}. \tag{39}$$

This can be written as a function of the material parameters using a simplified Mohr Coulomb criterion (Eq. 11) for the 1D case (Appendix B):

$$\sigma_{II} + \sin\phi\,\sigma_I = \frac{1 + 2\sin\phi}{3}\sigma_{yy} < c, \tag{40}$$

where $\nu = 1/3$ was used. Substituting $\sigma_{yy}$ from Eq. 39 into Eq. 40, the yield criterion can be written in terms of the wind forcing and the material parameters:

$$\tau_a < \frac{3c}{F_{down}(2\sin\phi + 1)}, \tag{41}$$





Using our cohesion estimates ($15 kPa < c < 21 kPa$) and angles of internal friction in the range of observations (30 and $45°$,
this would suggest stable bands of landfast ice of extent $F_{down} \sim 100$ km should be sustainable. This is much larger than those
observed in the Arctic, where bands of landfast ice rarely exceed a few tens of kilometers unless anchor points are provided by
stamukhi (Mahoney et al., 2014). This discrepancy highlights that neglected processes such as waves and tides are important
in shaping the landfast ice cover, and may prevent such landfast extent from occuring.

### 3.2.4 Compressive strength criterion

Including a compressive strength criterion ($\sigma_I > \sigma_c$) can modify the upwind fracture event (stage D) by the development
of compression fractures along the upwind coast of the islands. The compression strength only affects the simulation if the
compressive stress upwind of the islands exceeds the compressive strength.

As for the downwind case, the critical wind stress for the development of a compressive fracture can be approximated using
the 1D version of the momentum equation. The maximum normal stress at the upwind coast of the islands is:

$$\sigma_{yy} = \tau_a F_{up}. \tag{42}$$

where $F_{up}$ is the distance between the top boundary of the domain and the upwind coasts of the islands (see Fig. 4). In the
ideal case, the compression strength criterion is:

$$\sigma_I = \frac{(1+\nu)\sigma_{yy}}{2} > \sigma_c. \tag{43}$$

The compression criterion can thus be written as a function of the wind forcing, as:

$$\tau_a > \frac{2\sigma_c}{(1+\nu)F_{up}}. \tag{44}$$

Whether the ice will fail in shear (Mohr-Coulomb criterion) or in compression can be evaluated by substituting $\tau_a$ in Eq. (37)
by Eq. 44, yielding the criterion:

$$\frac{(1+\nu)F_{up}c}{W} > \sigma_c. \tag{45}$$

If this condition is met, the compression strength criterion does not influence the simulation, and the upwind shear fracture
lines develop as in the control simulation (Fig. 17a). If the left hand side term of Eq. 45 is much smaller that $\sigma_c$, compression
fracture occurs before the ice bridge break up and a ridge forms along the upwind coastlines, propagating in the channel
entrance while the ice in the channel remains landfast (Fig. 17b). If the terms are of similar order, the decohesion of the ice
bridge and the compression fractures are initiated simultaneously: compression fracture occurs along the upwind coastlines but
not in the channel entrance, as the ice both in and upwind of the channel starts to drift (Fig. 17c).

## 4 Conclusions

The MEB rheology was implemented on the Eulerian, Finite Difference numerical framework of the McGill sea ice model. We
show that the discretized Maxwell stress-strain relationship can be written in a form that resembles that of the VP model, with



an additional memory term. The MEB rheology is then simply implemented by redefining the VP viscous coefficients in terms of the MEB parameters and by adding the damage parameterization in a separate module. To our knowledge, this is the first time the MEB rheology is implemented on the same framework of a VP or EVP model. This will allow direct comparison of these models using the same numerical platform in future work.

In idealised ice bridge simulations, we show that the damage parameterization allows the ice fractures in the MEB model to propagate over large distances at short time scales. This process relies on the memory of the past deformations included in the model which cause a concentration of stresses close the preexisting damage. We also show that while the choice of yield curve influences the localisation and orientation of the ice fractures, the angles of fracture propagation differ from those expected in a granular material such as sea ice. This is due to the simple stress correction scheme using a scalar damage parameter applied on all stress tensor components, such that the flow rule is determined by the stress state independently from the orientation at which the ice fracture is occuring. The angles of the fractures could be more physical by the use of a damage tensor, with a stress correction path derived from a physical flow rule. This is left as a future model development.

The stress correction scheme in the damage parameterization (Rampal et al., 2015) is also found to cause a problematic propagation of numerical errors in the stress memory terms. The magnitude of the error propagation depends on the magnitude of the compressive stress associated with the ice failure. These errors accumulate in the memory term at each fracture event, creating numerical artifacts that dominate the solutions over time. We argue that this weakness of the damage parameterization should be treated as a numerical issue. Note that these errors are hardly detectable when using material heterogeneity (Dansereau et al., 2016) or realistic boundaries, in which cases the problem is no longer symmetric. A possible solution to this problem would be to use a non-linear yield curve which converges to the Tresca criterion for large compressive stresses (e.g. the yield criterion of Schreyer et al. 2006). The use of a damage tensor and a different stress correction scheme would also solve this problem.

The simulated break up of the landfast ice bridge occurs with two main fracture events. First, an ice arch develops at the downwind end of the channel, shaping the edge of the ice bridge in the channel. This ice arch forms in all simulations and is stable in shape as long as the ice bridge remains in place, with a curvature that increases for smaller angles of internal friction. Second, shear fractures are formed at the upwind end of the channel, resulting in the decohesion of the channel ice bridge and in the formation of landfast wedges upwind of the islands. The angle of the landfast ice wedges depend on the angle of internal friction. Based on the simulation results, we determined that the parameterized cohesion most consistent to the observed ice bridges in the Arctic are in the range of 15-21 kPa, that is in the lower range of previous estimate. This result is consistent with the fact that only the wind forcing is considered in these idealized simulations, other forcings such as tides, ocean currents and thermal forces are likely acting in conjunction with the wind forcing.

Based on these results, these are our recommendations for using the MEB model:

– A very low solution residual tolerance $\epsilon_{res} < 10^{-10}$ should be use to limit the accumulation of errors associated to the correction scheme.

– The cohesion should be limited to $c < 21$ kPa.



- The cohesion should be written as a function of the ice condition (i.e as $E$, pr $P^*$ in the VP models). This should prevent the formation of unrealistically large ridges.

- An alternative stress correction scheme should be developed to limit the accumulation of errors in the stress memory
terms.

*Code availability.* Our sea-ice model code and outputs are available upon request.

**Appendix A: Damage factor $\Psi$**

Let $\sigma'_I$ and $\sigma'_{II}$ be the stress invariant at time level n before the correction is applied, and $\sigma_{If}$ and $\sigma_{IIf}$ the corrected stress invariant lying on the yield curve. Following Bouillon and Rampal (2015) we use a damage factor $\Psi$ ($0 < \Psi < 1$) to reduce the
elastic stiffness and bring the stress state onto the yield curve. I.e. :

$$\sigma_{If} = \Psi\sigma'_I \quad ; \quad \sigma_{IIf} = \Psi\sigma'_{II}. \tag{A1}$$

Substituting these relations into the Morh Coulomb criterion ($\sigma_{IIf} + \mu\sigma_{If} = c$) we solve for $\Psi$:

$$\Psi = \frac{c}{\sigma'_{II} + \mu\sigma'_I}. \tag{A2}$$

Note that this relation implies that the stress correction is done following a line from the stress state ($\sigma'_I$,$\sigma'_{II}$) to the origin (see
Fig. 2). This scheme stems from the use of a single damage factor applied on the elastic stiffness, which is linear to each of the stress components, i.e., the correction is applied equally to each stress component. Other paths could be used for the correction (e.g. following a vertical or horizontal line), but would require the use of a different stress factor for each component of the stress tensor (i.e. using a damage tensor, rather than a damage parameter). This could be used to cure the error propagation problem when large compressive stresses are present (see Appendix C).

**Appendix B: Analytical solutions of the 1D momentum equation**

Considering an infinite channel of landfast ice ($\mathbf{u} = 0$) along the y-direction with wind forcing $\tau_a = \tau_{ay}$ and water stress $\tau_w = 0$, we write the 1D steady state momentum equation as:

$$\frac{\partial\sigma_{xy}}{\partial x} + \tau_{ay} = 0, \tag{B1}$$

where we have neglected the $\partial/\partial y$ terms. In this case, the normal stress is zero in the entire channel and the stress invariants
are $\sigma_I = 0$, $\sigma_{II} = \sigma_{xy}$. The shear stress at any arbitrary point $x$ across the channel can be determined by integrating Eq. B1 from the channel center ($x = 0$) to x :

$$\sigma_{xy} = -\tau_{ay}x. \tag{B2}$$





By symmetry, the maximum shear stresses in the channel are located at the channel walls, at $x = \pm\frac{W}{2}$ where $W$ is the width of the channel. The maximum shear stress invariant on the channel walls is then:

$$\sigma_{II} = \frac{W\tau_{ay}}{2}. \tag{B3}$$


Similarly, we find the analytical solution for the normal stresses in a band of landfast ice with width $L_y$ along an infinite coastline running in the $x$ direction, with a wind forcing $\tau = \tau_{ay}$ and water drag $\tau_w = 0$, by integrating the 1D momemtum equation in which the $\partial/\partial x$ terms are neglected. I.e. :

$$\frac{\partial \sigma_{yy}}{\partial y} + \tau_{ay} = 0, \tag{B4}$$


$$\sigma_{yy} = -\tau_{ax}y. \tag{B5}$$

Placing the landfast ice edge (where $\sigma_{yy} = 0$) at $y = 0$, the largest compressive stresses will be located along the coast, at $y = -L_y$. Note that in this case, shear stress is zero in the entire land-fast ice and the stress invariants are function of both $\sigma_{xx}$ and $\sigma_{yy}$:

$$\sigma_{yy} = EC_1\epsilon_{yy} \tag{B6}$$


$$\sigma_{xx} = EC_2\epsilon_{yy} = \nu\sigma_{yy} \tag{B7}$$

$$\sigma_I = \frac{\sigma_{xx} + \sigma_{yy}}{2} = \frac{(1+\nu)\sigma_{yy}}{2} \tag{B8}$$


$$\sigma_{II} = \sqrt{(\frac{\sigma_{yy} - \sigma_{xx}}{2})^2} = \frac{(1-\nu)\sigma_{yy}}{2} \tag{B9}$$

This allows to write the Mohr-Coulomb criterion in terms of $\sigma_{yy}$:

$$\sigma_{II} + \sin\phi\,\sigma_I = \frac{1 + 2\sin\phi}{3}\sigma_{yy} < c, \tag{B10}$$

## Appendix C: Error propagation analysis

The error $\delta F$ associated with a function $F(X, Y, Z, ...)$ with uncertainties $(\delta x, \delta y, \delta z, ...)$ is given by:

$$\delta F = \sqrt{(\frac{\partial F}{\partial X})^2 \delta x^2 + (\frac{\partial F}{\partial Y})^2 \delta y^2 + (\frac{\partial F}{\partial Z})^2 \delta z^2 + ....} \tag{C1}$$




In the damage parameterization, the components of the corrected stress tensor used as the memory terms ($\sigma_{ijM}$) can be written in terms of the uncorrected stress tensor ($\sigma'_{ij}$) and the damage factor $\Psi$ (Eq. 13):

$$\sigma_{ijM} = \psi \sigma'_{ij}. \tag{C2}$$

Using Eq. A2, this can be rewritten in terms of the uncorrected stress invariants ($\sigma'_I$, $\sigma'_{II}$):

$$\sigma_{ijM}(\sigma'_{ij}, \sigma'_I, \sigma'_{II}) = \frac{c\, \sigma'_{ij}}{\sigma'_{II} + \mu \sigma'_I} \tag{C3}$$

Assuming that the model has converged to a solution within an error on the stresses $\delta\sigma'_{ij} = \epsilon \sigma'_{ij}$, $\delta\sigma'_I = \epsilon \sigma'_I$, $\delta\sigma'_{II} = \epsilon \sigma'_{II}$, where $\epsilon$ is a small number, the model convergence error propagates on the stress memory with an error of :

$$\delta\sigma_{ijM} = \sqrt{\Big(\frac{\partial \sigma_{ijM}}{\partial \sigma'_{ij}}\Big)^2 \delta\sigma'^2_{ij} + \Big(\frac{\partial \sigma_{ijM}}{\partial \sigma'_I}\Big)^2 \delta\sigma'^2_I + \Big(\frac{\partial \sigma_{ijM}}{\partial \sigma'_{II}}\Big)^2 \delta\sigma'^2_{II}}. \tag{C4}$$

Substituting ($\delta\sigma'_{ij}$, $\delta\sigma'_I$, $\delta\sigma'_{II}$) for $\epsilon$ and using Eq. C3, we obtain:

$$\delta\sigma_{ijM} = \sqrt{\frac{c^2}{(\sigma'_{II} + \mu \sigma'_I)^2} \epsilon^2 \sigma'^2_{ij} + \frac{c^2 \sigma'^2 \mu^2}{(\sigma'_{II} + \mu \sigma'_I)^4} \epsilon^2 \sigma'^2_I + \frac{c^2 \sigma'^2}{(\sigma'_{II} + \mu \sigma'_I)^4} \epsilon^2 \sigma'^2_{II}}, \tag{C5}$$

or:

$$\delta\sigma_{ijM} = \epsilon \sigma_{ijM} \sqrt{1 + \frac{\sigma'^2_{II} + \mu^2 \sigma'^2_I}{(\sigma'_{II} + \mu \sigma'_I)^2}}. \tag{C6}$$

Assuming that error on the stress memory components ($\epsilon_M$) has the form $\delta\sigma_{ijM} = \epsilon_M \sigma_{ijM}$, we can express the relative error
of the stress memory components as a function of of the stress invariants as :

$$\epsilon_M = \epsilon \sqrt{1 + R} \tag{C7}$$

where

$$R = \frac{\sigma'^2_{II} + \mu^2 \sigma'^2_I}{(\sigma'_{II} + \mu \sigma'_I)^2} \tag{C8}$$

*Author contributions.* M. Plante coded the model, ran all the simulations, analysed results and led the writing of the manuscript. B. Tremblay
participated in regular discussions during the course of the work and edited the manuscript. M. Losch participated in regular discussions during M. Plante research stays in Germany and edited the manuscript. J-F. Lemieux participated in the implementation of the MEB rheology and provided edits on the manuscript.

*Competing interests.* The authors declare that they have no conflict of interest.





*Acknowledgements.* Our sea-ice model code and outputs are available upon request. Mathieu Plante would like to thank the Fonds de

recherche du Québec – Nature et technologies (FRQNT) for financial support received during the course of this work. Bruno Tremblay is grateful for support from the Natural Science and Engineering and Research Council (NSERC) Discovery Program and the Office of Naval Research (N000141110977). This work is a contribution to the research program of Québec-Océan and of the ArcTrain International Training Program. We acknowledge the use of imagery from the NASA Worldview application (https://worldview.earthdata.nasa.gov), part of the NASA Earth Observing System Data and Information System (EOSDIS).



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





**Table 1.** Material strength parameters from observations

| Parameter | Reference | Parameter | Value |
| --- | --- | --- | --- |
| Young Modulus | Langleben (1962) | E | $6.5 - 10$ GPa |
| | Weeks and Assur (1967) | | $1 - 9$ GPa |
| | Tabata (1955) | | $7 - 18$ GPa |
| Poisson ratio | Weeks and Assur (1967) | $\nu$ | 0.33 - 0.4 |
| Viscosity | Tabata (1955) | $\eta_0$ | $0.6 - 2.4 \times 10^{12}$ kgm$^{-1}$s$^{-1}$ |
| Viscous relaxation time | Tabata (1955) [a] | $\lambda_0$ | $14 - 40$ min |
| | Weeks and Assur (1967)[a] | | $28 - 32$ min |
| | Sukhorukov (1996)[a] | | 66 h |
| | Hata and Tremblay (2015b) | | $10^5$ s |
| Angle of internal friction | Schulson et al. (2006) | $\phi$ | $\sim 42°$ |
| | Weiss et al. (2007) | | $\sim 44°$ |
| Compressive strength | Weiss et al. (2007) | $\sigma_c$ | 50 kPa |
| | Tremblay and Hakakian (2006)[b] | | 30 - 100 kPa |
| | Tucker and Perovich (1992)[c] | | 30 kPa |
| | Richter-Menge et al. (2002)[c] | | 30-50 kPa |
| | Richter-Menge and Elder (1998)[c] | | 100-200 kPa |
| Tensile strength | Weiss et al. (2007) | $\sigma_t$ | 50 kPa |
| | Tremblay and Hakakian (2006)[b] | | 25-30 kPa |
| | Tucker and Perovich (1992)[c] | | 30 kPa |
| | Richter-Menge and Elder (1998)[c] | | 50 kPa |
| Cohesion | Weiss et al. (2007) | $c$ | 40 kPa |

[a] Form small scale measurements in the field.

[b] Estimate from satellite observations.

[c] Observed peak stresses.





**Table 2.** Default Model Parameters

| Parameter | Definition | Value |
| --- | --- | --- |
| $\Delta x$ | Spatial resolution | 2 km |
| $\Delta t$ | Time step | 2 s |
| $T_d$ | Damage time scale | 2 s |
| Y | Young Modulus | 1 GPa |
| $\nu$ | Poisson ratio | 0.3 |
| $\lambda_0$ | Viscous relaxation time | $10^7$ s |
| $\phi$ | Angle of internal friction | $45°$ |
| c | Cohesion | 20 kN/m |
| $\sigma_t$ | Isotropic tensile strength | 28 kN/m |
| $\sigma_c$ | Isotropic compressive strength | 50 kN/m |
| $\rho_a$ | Air density | 1.3 kg/m$^3$ |
| $\rho_i$ | Sea ice density | $9.0 \times 10^2$ kg/m$^3$ |
| $\rho_w$ | Sea water density | $1.026 \times 10^3$ kg/m$^3$ |
| $C_{da}$ | Air drag coefficient | $1.2 \times 10^{-3}$ |
| $C_{dw}$ | Water drag coefficient | $5.5 \times 10^{-3}$ |





**Table 3.** Material properties used in sea ice models (VP,EVP and MEB)

| Parameter | Reference | Parameter | Value |
|---|---|---|---|
| Young Modulus | Hunke (2001) | $E = \zeta/T$ | 1060 GPa |
| | Bouillon and Rampal (2015) | $Y$ | 9 GPa |
| | Dansereau et al. (2016) | $E_0$ | 0.585 GPa |
| | Sulsky and Peterson (2011) | $E$ | 1 MPa |
| | Tran et al. (2015) | $E$ | 1 MPa |
| Maximum Viscosity | Ólason (2016) | $\zeta_{max}$ | $378 \times 10^{15}$ kg/s |
| | Dansereau et al. (2016)[a] | $\eta_0 = 10^7 E_0$ | $5.85 \times 10^{15}$ kg/ms |
| | Hunke (2001) | $\zeta_{max}$ | $1375 \times 10^{12}$ kg/s |
| | Tremblay and Mysak (1997) | $\eta_{max}$ | $1 \times 10^{12}$ kg/s |
| | Hibler (1979) | $\zeta_{max}$ | $125 \times 10^9$ kg/s |
| | Dumont et al. (2008) | $\zeta_{max}$ | $4 \times 10^8$ |
| Compressive strength | Tran et al. (2015) | $f'_c$ | 125 kPa |
| | Sulsky and Peterson (2011) | $f'_c$ | 125 kPa |
| | Lemieux et al. (2016)[a] | $P_p$ | 100 kPa |
| | Ólason (2016) | $p^*$ | 40 kPa |
| | Dansereau et al. (2016) | $\sigma_c$ | 48 - 96 kPa |
| | Hunke (2001)[a] | $P$ | $27.5 \times 10^4$ kPa |
| | Dumont et al. (2008) | $P^*$ | 27.5 kPa |
| | Bouillon and Rampal (2015) | $\sigma_{Nmin} = -\frac{5}{2}c$ | $1.25 - 20$ kPa |
| | Tremblay and Mysak (1997) | $P_{max}$ | 7 kPa |
| | Hibler (1979) | $P^*$ | 5.0 kPa |





**Table 3.** Table 3 continued

| Parameter | Reference | Parameter | Value |
|---|---|---|---|
| Shear strength : | Hibler (1979) | $e$ | 2 |
| | Hunke (2001) | $e$ | 2 |
| | Dumont et al. (2008) | $e$ | $1.2 - 1.6$ |
| | Lemieux et al. (2016) | $e$ | $1.4 - 1.6$ |
| | Ólason (2016) | $e$ | $1.3 - 2.1$ |
| | Dansereau et al. (2016) | $C$ | 25 - 50 kPa |
| | Ólason (2016)** | $\sigma_{uc}$ | $16 - 22$ kPa |
| | Tran et al. (2015) | $\tau_{sf}$ | $15 - 75$ kPa |
| | Sulsky and Peterson (2011) | $\tau_{sf}$ | 15 kPa |
| | Bouillon and Rampal (2015) | $c$ | $0.5 - 8$ kPa |
| Tensile strength | Ólason (2016)[b] | $Pk_t$ | $3.4 - 5$ kPa |
| | Lemieux et al. (2016) | $k_t P_p$ | $10 - 20$ kPa |
| | Beatty and Holland (2010) | $k_t$ | 27.5 kPa |
| | Dansereau et al. (2016) | $\sigma_t = 0.27\sigma_c$ | $12.96 - 25.92$ kPa |
| | Tran et al. (2015) | $\tau_{nf}$ | 25 kPa |
| | Sulsky and Peterson (2011) | $\tau_{nf}$ | 25 kPa |
| | Bouillon and Rampal (2015) | $\sigma_{Nmax} = \frac{5}{4}c$ | $0.6 - 10$ kPa |

[a]for 1m thick ice

[b]Using the Mohr-Coulomb curve with $\phi = 45°$



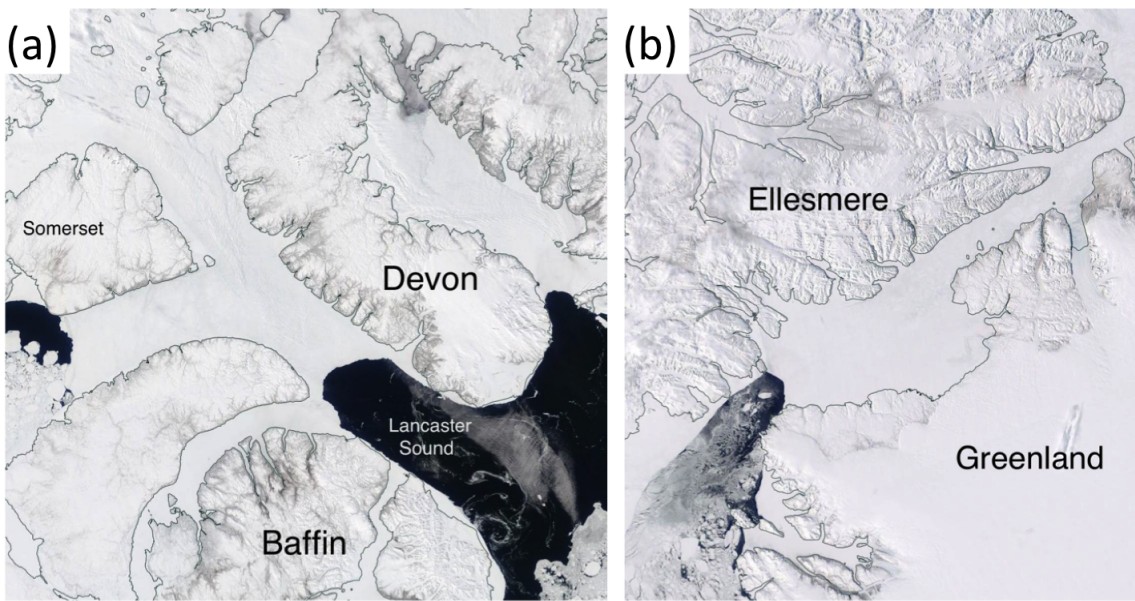

**Figure 1.** NASA Worldview images of landfast ice arches, from Moderate Resolution Imaging Spectroradiometer (MODIS) Corrected

Reflectance imagery (True Color). a) Multiple ice arches in the Northwest Passage region, on June 7th 2018. b) Nares Strait ice arch, on May

1st 2018.



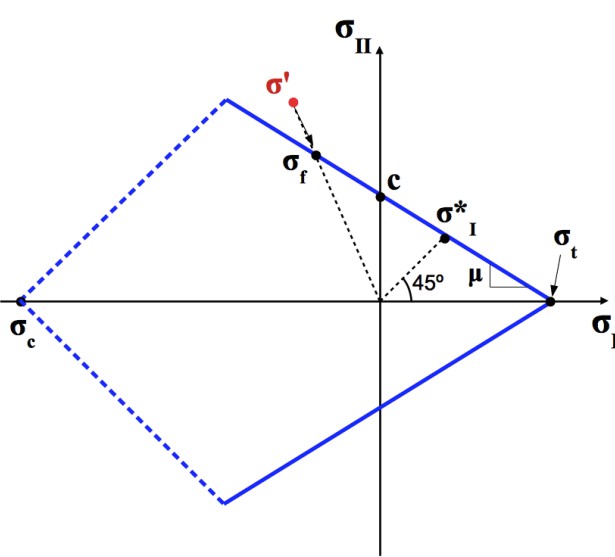

**Figure 2.** Mohr-Coulomb yield criterion in stress invariant space ($\sigma_I$, $\sigma_{II}$) with the mechanical strength parameters: compressive strength ($\sigma_c$), cohesion ($c$), coefficient of internal friction ($\mu = \sin\phi$, $\phi$ being the angle of internal friction), isotropic tensile strength ($\sigma_t$) and uniaxial tensile strength ($\sigma_I^*$, where the second principal stress invariant $\sigma_2$ is zero, or $\sigma_I = \sigma_{II} = \sigma_I^*$). The stress before and after the correction (see Eq. 13) is denoted by $\sigma'$, and $\sigma_f$ respectively. The correction from $\sigma'$ to $\sigma_f$ is done following a line going through the origin.



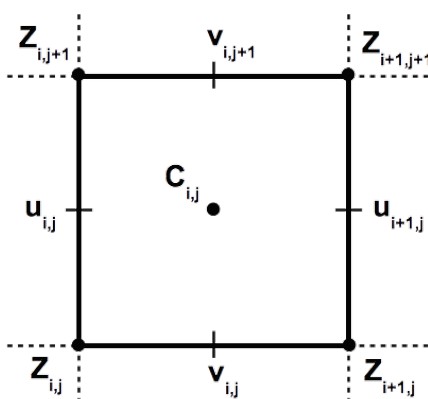

**Figure 3.** Location of the scalars ($C_{i,j}$) and vector components ($u_{i,j}, v_{i,j}$) on the Arakawa C-grid. The normal and shear stress components used in the memory term are located center ($C_{i,j}$) and nodes ($Z_{i,j}$) respectively.





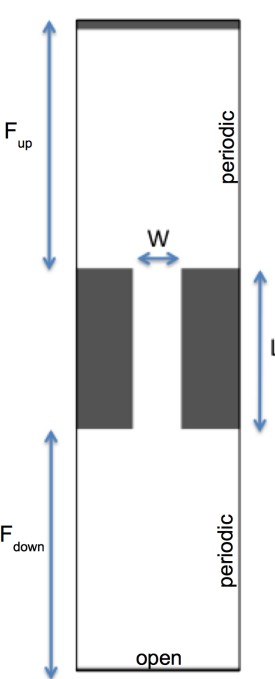

**Figure 4.** Idealized domain with a solid wall to the north, open boundary to the south and periodic boundaries to the East and West. The channel has a width W, length L and fetch $F_{up}$ and $F_{down}$ in the upwind and downwind basins respectively.



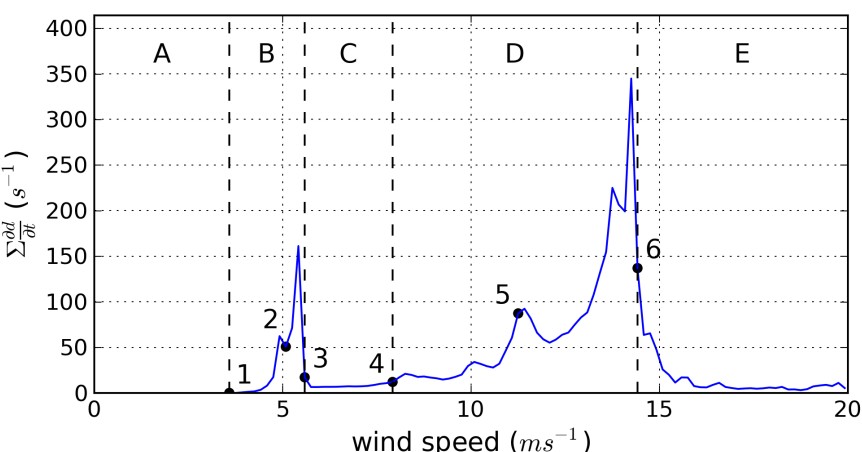

**Figure 5.** Top pannel: time series of the domain integrated brittle fracture activity ($\partial d/\partial t$) for the control run simulation. Dashed lines indicate the beginning and end of the simulation phases (A,B,C,D,E). Numbers indicate the location of the damage field in Fig. 6 and 12.





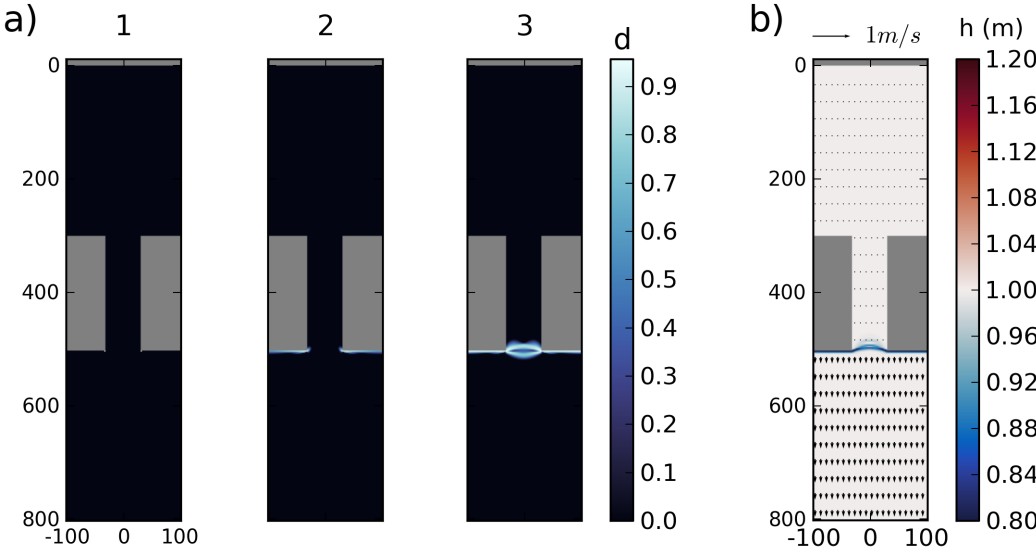

**Figure 6.** a) Damage field during the formation of the downwind ice arch, at points indicated in Fig. 5. b) Sea ice thickness and drift following the formation of the downwind ice arch, while the ice bridge remains stable (Phase C)


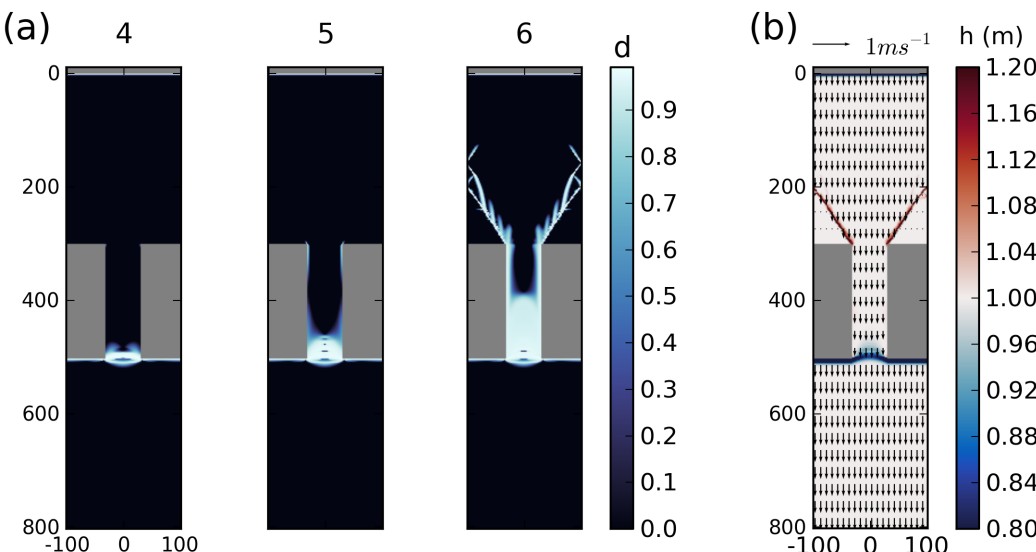

**Figure 7.** a) Damage field during ice formation of upwind lines of fracture, at points indicated in Fig. 5. b) Sea ice thickness and drift following the ice bridge collapse (Phase E)


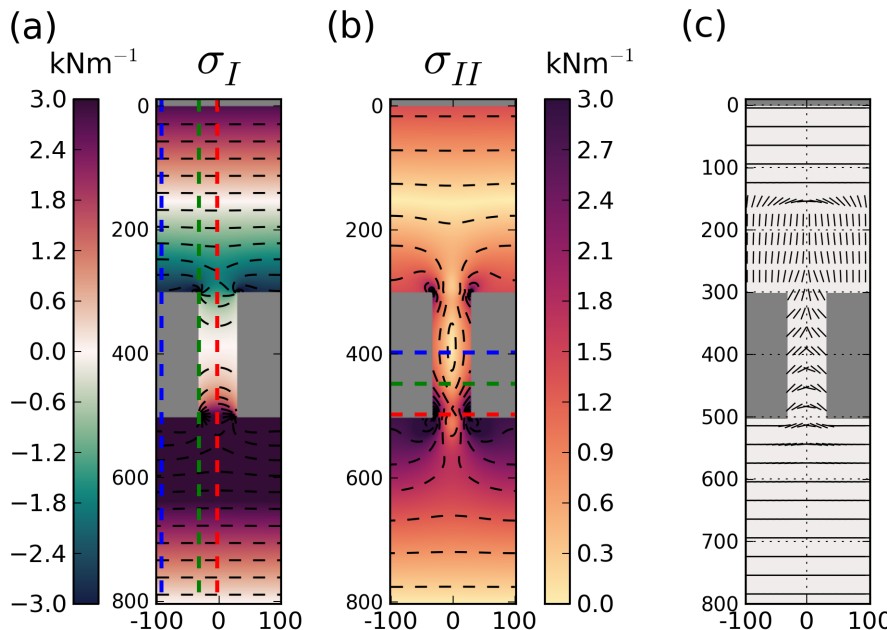

**Figure 8.** Stress fields in landfast ice during Phase A. a) Normal stress invariant ($\sigma_I$), with colored lines to indicate the vertical transects used in Fig. 9, b) shear stress invariant ($\sigma_{II}$), with colored lines to indicate the vertical transects used in Fig. 9, c) orientation of the second principal stress componant.



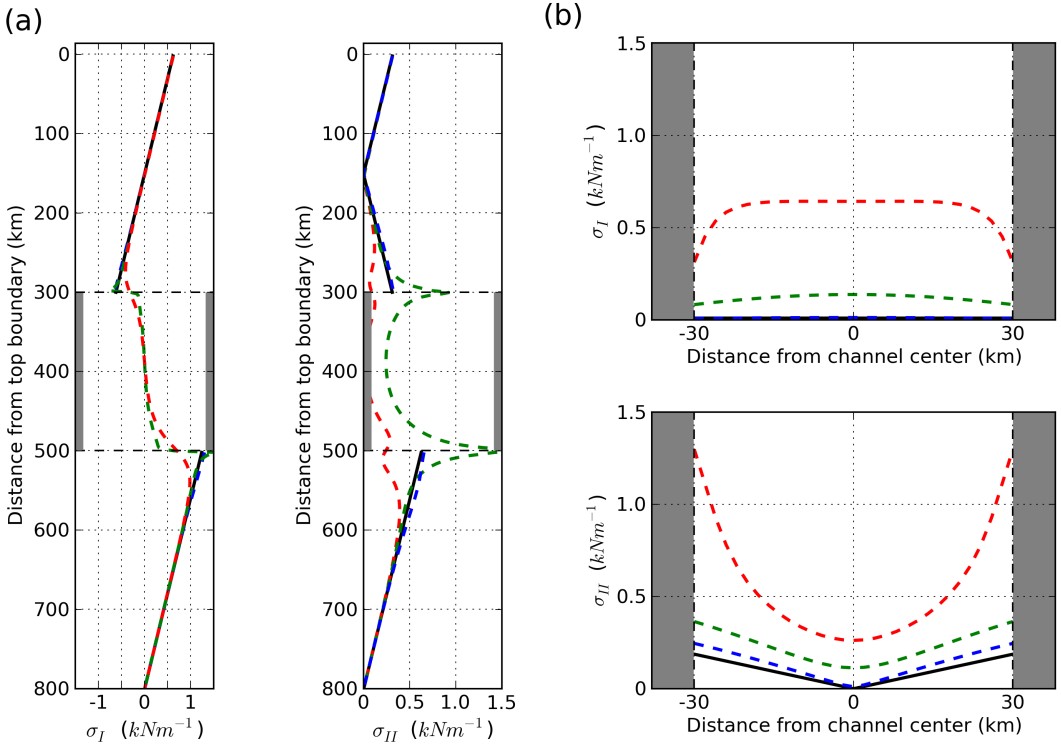

**Figure 9.** Stress invariants ($\sigma_I$, $\sigma_{II}$) along the transects of corresponding colors in Fig. 8: a) transects running along the y-direction and b) transects running along the x-direction. Black solid lines indicate the analytic solutions. Grey area indicate the position of the islands.

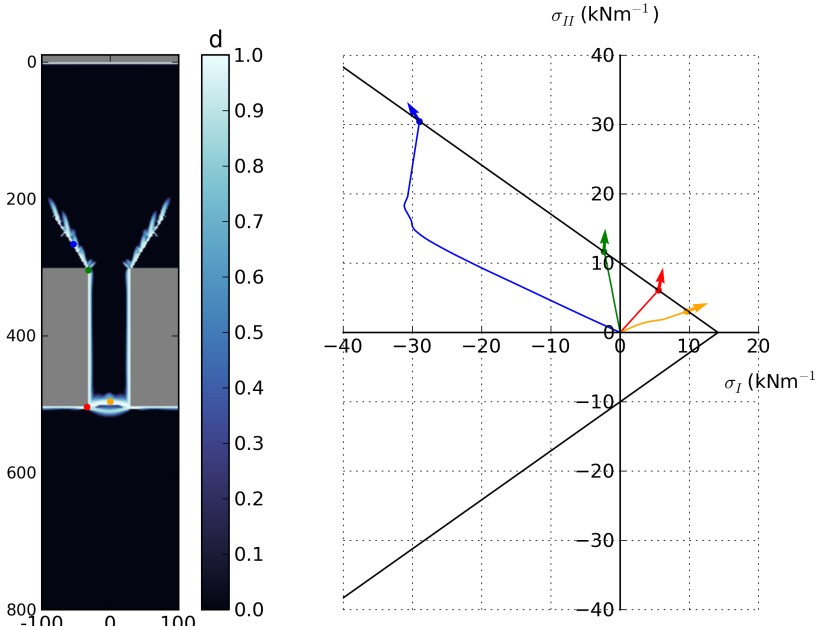

**Figure 10.** Left: Location of four points associated to the first fracture in the domain (red), the center of the ice arch (orange), the fracture at the upstream channel corners (green) and along the upstream shear fracture line (blue). Right : stress state and orientation of the strain rate tensor at the 4 indicated points they reach the yield curve. The Mohr Coulomb criterion is in black. Colored line indicate the path of the stress state at each points prior to the fracture.





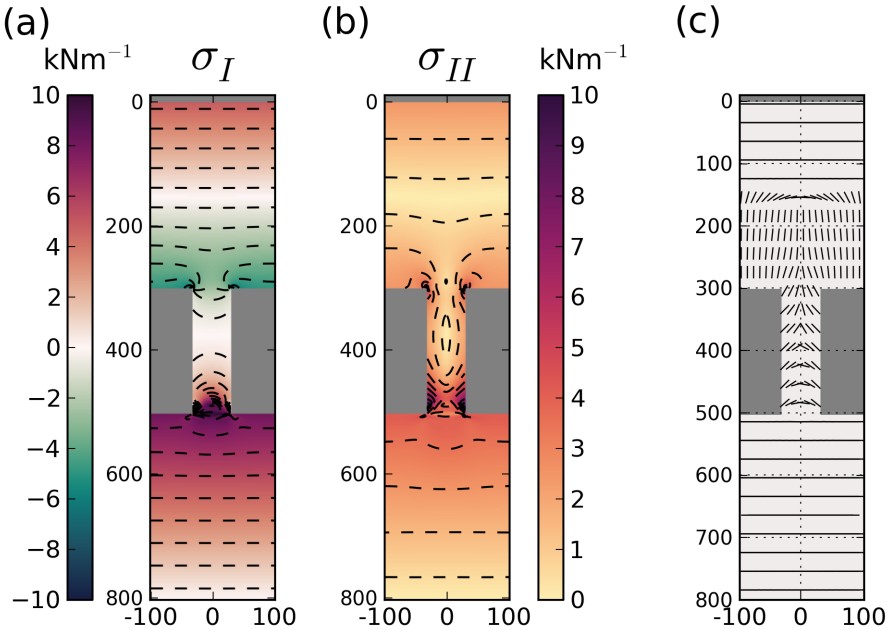

**Figure 11.** Stress fields during Phase C. a) Normal stress invariant ($\sigma_I$), with colored lines to indicate the vertical transects used in Fig. 9, b) shear stress invariant ($\sigma_I$), with colored lines to indicate the vertical transects used in Fig. 9, c) orientation of the second principal stress componant.





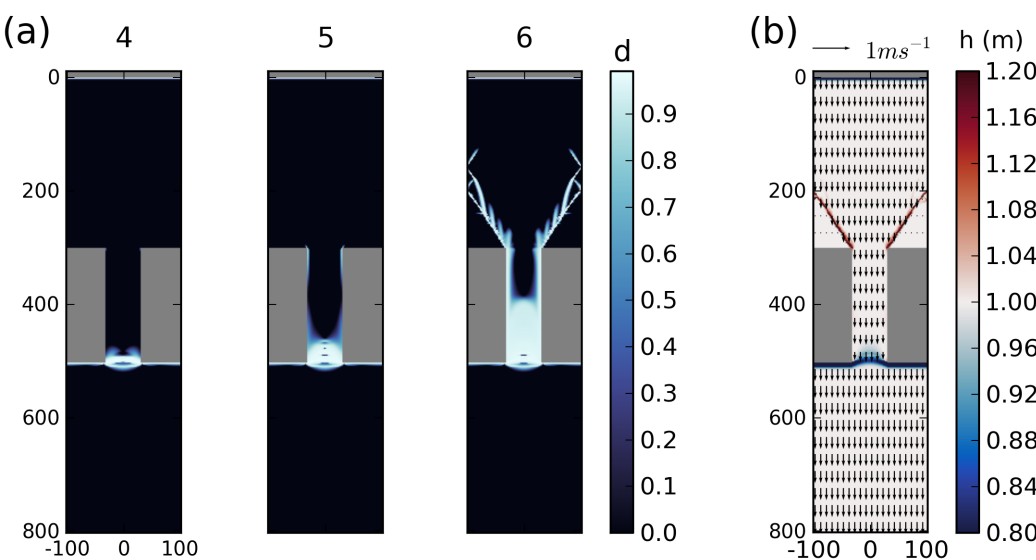

**Figure 12.** a) Damage field during ice formation of upwind lines of fracture, at points indicated in Fig. 5. b) Sea ice thickness and drift following the ice bridge collapse (Phase E)


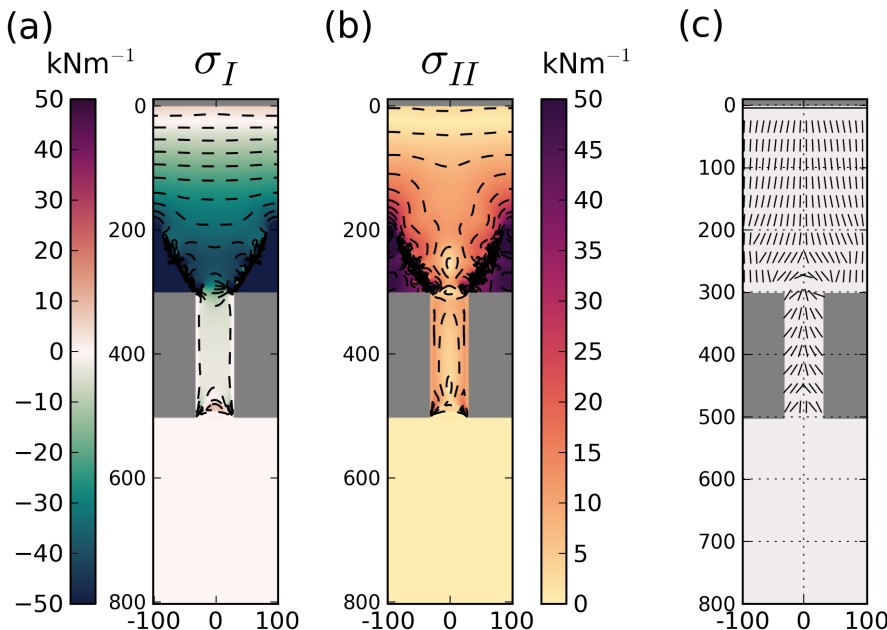

**Figure 13.** Stress fields during Phase E. a) Normal stress invariant ($\sigma_I$), with colored lines to indicate the vertical transects used in Fig. 9, b) shear stress invariant ($\sigma_I$), with colored lines to indicate the vertical transects used in Fig. 9, c) orientation of the second principal stress component.

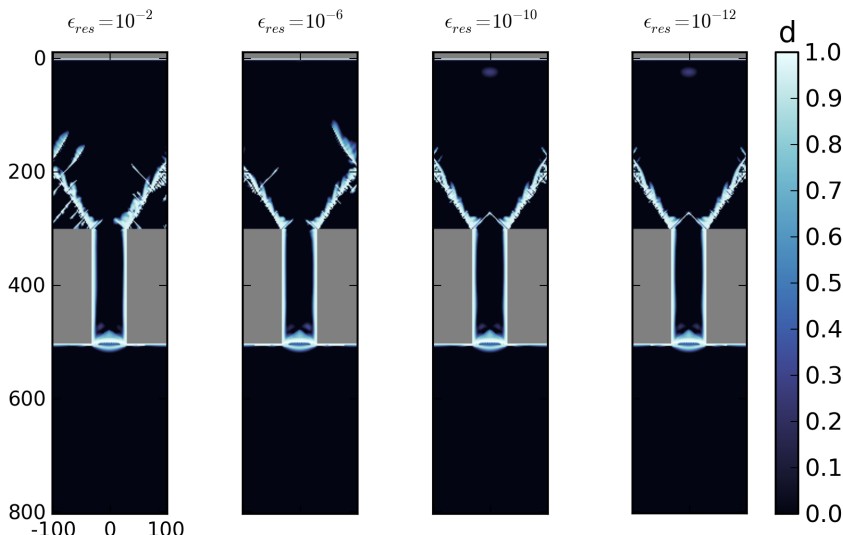

**Figure 14.** Damage fields during Phase E for different solution residual tolerance. Left: $\epsilon_{res} = 10^{-6}$, right: $\epsilon_{res} = 10^{-10}$



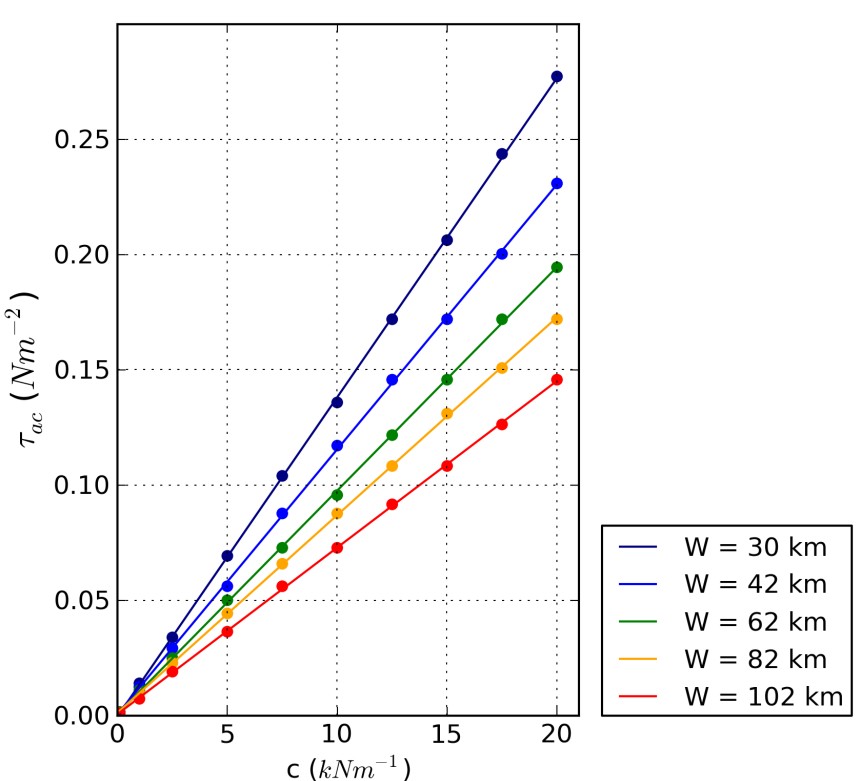

**Figure 15.** Critical wind forcing associated with the second fracture event (stage D) as a function of cohesion and channel width. The graph

on the right shows the slope between the critical forcing and the cohesion as a function of the channel width.





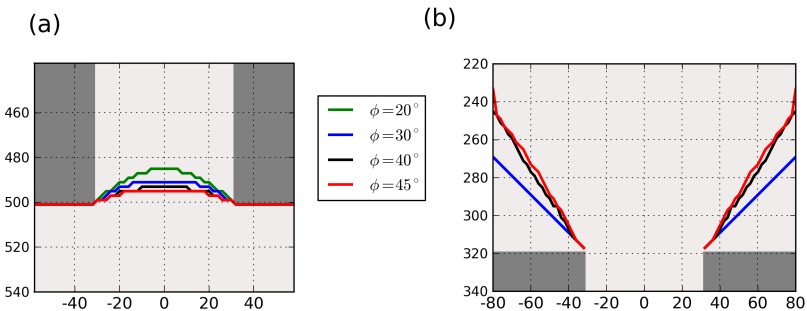

**Figure 16.** Shape of the lines of fracture using different angles of internal friction: a) for the downwind ice arches and b) for the upwind lines of fracture (the green and blue lines are superposed).



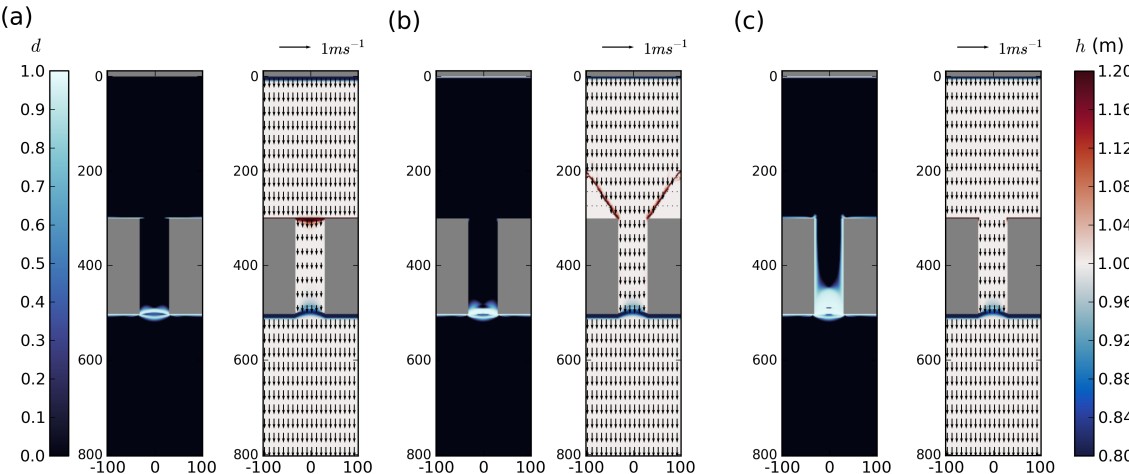

**Figure 17.** Spatial distribution of the damage field at the end of stage D (left) and the sea ice thickness and velocity fields at the end of the simulation (right). For different compressive strength criterion: a) $\sigma_c = 5.0$ kN/m, b) with $\sigma_c = 25.0$ kN/m and c) with $\sigma_c = 100.0$ kN/m.