# Peer review of "Landfast sea ice material properties derived from ice bridge simulations using the Maxwell Elasto-Brittle rheology"

_The Cryosphere, 2019_

## Referee Comment (RC1) · Anonymous Referee #1 · 17 Nov 2019

The manuscript named "The material properties of ice bridges in the Maxwell Elasto-Brittle rheology" test the MEB sea-ice rheology in a traditional finite difference framework. The aim is to investigate the damage parameterisation. This is achieved with an idealized model setup of a channel that is narrow in the middle an wider In the two ends. I think that this is a very relevant to study the damage parameterisation as this (at least in my opinion) is important for new developments within sea-ice dynamics. The manuscript is in general well written and therefore fairly easy to read.

I could wish for a better organization of the figures. In addition, the result section would be much easier to read if the figures were numbered in the order they are referenced. At last I think that the simplifications made in terms of zero ocean currents/sea surface tilt may have bigger impact, especially in some of the examples mentioned within the

Canadian archipelago, where tides are significant. This is not necessary to include within this study but a better discussion of the limitation would be nice.

The results in general seems less impressive than other studies using the MEB rheology. A discussion of the performance of these would also be nice. Many of these points are mentioned but I think that it would be beneficial to collect these in a discussion and maybe spend a few more words.

Mayor revisions:

I think that the focus of the abstract is a bit off and I would like this to be revised. It is not that important that the framework of the sea-ice model originally was build for VP dynamics as this is not mentioned in the manuscript. The eulerian/lagrangian implementation is more relevant. I would like the abstract to include line 77-82 as these fit well into a summary/abstract and less into the introduction Discussions are scattered around in the manuscript. I would like a collected discussion.

- What improvements/limitations are there when the framework moves from a Lagrangian to an Eulerian approch

The study limits the effects of the ocean(eg tides) by neglecting it. For an idealized study like this it is fine just to look at the wind. But in terms of comparisons with real data then this restricts the value og the study. Tides are very high from Kanes Basin and southwards. This is an important factor when the stability of the last fast ice is to be considered and compared with real life. This is briefly mentioned on line 500, however I would like a bigger discussion of this.

Figures are very inconsistent when labeling. These should be changed. I have suggested updates to almost all of the figures. These comments are in the minor correction/technical correction part.

Minor details

Line 2- Please revise sentence. An example is provided: The effect of the material

parameters on ice arches in a numerical framework that includes both the Maxwell Elasto-Brittle (MEB) including a damage parameterization and the Viscous-Plastic (VP) dynamics.

Following lines after line 2: I assume that this is MEB but it is a little unclear

Line 20 I think that this should be reformulated. For instance, ice keel don't protect sea-ice from forcing. It creates a friction that resist the forcing. I would reformulate this

Line 35 Ice thickness anomaly is this in time or space? I guess that the influencing factor is the current ice thickness, spatial variation (anomaly).

Line 47: replace new rheology with new rheologies

Line 67. References to figures in other articles makes it hard to read. Please either add the location on figure 1 or add a map where this can be shown.

Line 77 to 82: This part would be well suited for conclusions and/or abstract. The introduction should be more overview of previous studies and an overview of what will be presented. Not results.

Line 95 Nares Strait has strong tides in the part near Baffin Bay, thus the ocean currents would most likely be an important factor especially before and probably after the fast ice region has formed. Therefore, this should be mentioned in the discussion. Coriolis is only zero when the ice is not moving. The discussion of the influence of the ocean is too small. Equation 6 ":" in the equation?. This is described in equation 8. This should be moved here (first place that it is used)

Line 119 lhs and rhs should be written without using a abbreviation.

Line 120- 124. These sentences are a bit hard to follow. Please revise.

Line 265 Figure 3 No need to show a ARAKAWA grid. This is a standard. I would remove the figure.

Line 317 Remove "-" 2 times

Line 404 In short the physical solution did not converge until the tolerance is lower than $10^{-10}$. How many iterations are required? Is this important for the computational time (how important?)

Line 447 Nature is a bit more complex than just wind. Orography ocean currents etc also play a role, thus values like the cohesion of sea ice should be smaller than 21 KN/m seems to be a very rough estimate based on parts of the momentum equation. Admitted wind is normally the dominant factor along with the resistance (internal strength)

Line 473 How does this compare with results from other MEB implementations.

Line 520: is the ":" suppose to be there?

Line 528: It would be very interesting to include this in a VP/EVP model.

Figure 5 text. Top panel? I can only see one panel in figure 5.

Figure 6. This figure should be labeled a through d instead of a1 2 3 b

Figure 7 Same as 7, Which points?

Figure 8: I would say colored dashed lines

Figure 10 Dots are very hard to see. It would be nice to increase the size of these.

Figure 11: Which colored lines? They are defined in figure 8. Are they the same?

Figure 12: I would still label these a, b,c and d. Then add to the text.

Figure 13 which colored lines? Are they the same?

Figure 14. Please use a,b,c… References to the residual tolerance are not very easy. Left $10^{-6}$ and right $10^{-10}$ does not make sense.

Figure 17 Arrows are very hard to see.

Figures in general should be in order of them being mentioned in the text. For instance the result section seem to jump back and forth. I assume that when done with the review process they need to be inserted at appropriated places.
* * *

---

## Referee Comment (RC2) · Jérôme Weiss (Referee) · 17 Dec 2019

This manuscript presents an implementation of the recently developed Maxwell Elasto-Brittle rheology for sea ice mechanics within a finite-difference scheme, and the realization of idealized simulations of ice deformation and motion within a channel. This work is quite similar with what was done in [1], however with several differences:
- a finite-difference, instead of a finite-element, numerical scheme is used
- the numerical implementation of the MEB rheology is performed exploiting the code framework of a standard VP code.
- the initial condition is a purely homogeneous ice plate (constant thickness, constant elastic and strength properties), without any sort of initial disorder implemented
The effects of material properties and model parameters are then analyzed from a sensitivity analysis using strictly symmetric boundary conditions and geometry.
Overall, this manuscript is clearly written. Most of the results are consistent with [1], proving the robustness of the MEB rheology to adequately simulate sea ice damage, fracture, or strain localization. I am therefore rather favorable to a publication in The Cryosphere. There are however several mistakes, misunderstandings, or points to discuss more thoroughly (e.g., the flow rule and its relationship to the angle of fractures), which should be fixed before final acceptance. They are listed below:

1) Title: "The material properties of ice bridges…". This is strange, and unclear: bridges do not have "material properties". Maybe the authors wanted do say: "The effect of material properties on the simulation of ice bridges in the .." ?

2) p3, L65-73, as well as p14, L390-395: The authors argue at different places about the position of the ice bridges, up- or down-wind the channels. This argumentation is not very clear:
- to what extent the situation shown on fig. 1b is systematic ? To the best of my knowledge, ice bridges can take place at different positions along channels (as actually shown on fig. 1b), including along Nares strait (see e.g. fig. 1 of [1]), and I do not know a systematic, statistical analysis of that (but would be happy to learn about such analysis, if any).
- in [1], ice breakup occurs in several successive steps, at different locations along the Nares strait.
In conclusion, I am not sure that the unique observation of fig 1b can serve as a severe constrain on model parameterizations.

2) L92-93. Indeed, at the timescales involved in classical climate models (time step of several hours, as in the papers cited), the advection term can be neglected. This might not be true when considering a much smaller model time step (~s). What is the time step of the present simulations? Have you done a proper scale analysis taking that time step into account?

3) In section 2.1, ice thickening through mechanical redistribution when A=1 is not considered. Was such redistribution scheme implemented? If not, this would be a problem, as such scheme, even very simple, and coupled to the MEB rheology, was found to generate realistic ITD [1]. This would therefore likely affect all the discussion about ice ridges throughout the entire manuscript. If such scheme was implemented, please detail.

4) Equation (1): $C$ is not defined here, but much later (eq. (9)). Please modify.

5) L125-127: I would not agree and there seems to be a misunderstanding here: the elasto-brittle component of the MEB rheology is, by construction, associated with *small* (and reversible) deformations, while the Maxwell component deals with large (and irreversible) deformations. This is therefore fundamentally different from the VP model where the *plastic regime* is associated with irreversible strains but strain-rate independent stresses (while the Maxwell component of the MEB model is indeed *strain-rate strengthening* and not strain-rate independent). Please justify in physically-sound way or remove that sentence.

6) L134-137 and L222: At least in the case of (Tabata, 1955) and (Weeks and Assur, 1967), these authors discuss the creep of *bulk* saline ice, driven by viscoplasticity at the crystal scale. The concept introduced in the MEB rheology is fundamentally different: a linear viscous term is introduced to account for the cataclastic flow of highly damaged ice, and associated stress relaxations.

7) L153-154: This statement is wrong: plane stresses were considered in *Dansereau et al.*, 2015, 2016, 2017 and any implementation of the MEB rheology. This is indeed the correct assumption for thin plates. Note however that the impact of such assumption (plane stress vs plane strain) has little consequence on the global behavior. Note also that the constitutive equation present in the early newsletter *Dansereau et al., 2015* is that of the generic Maxwell model.

8) Section 2.2.2. The authors propose to close the damage envelope towards large compressive stresses using eq. (11). This is another difference with the initial MEB model [2]. In principle, I would say "Why not ?". However, several questions arise:
- What is the physical justification of such closure? At the lab scale, the failure envelope of columnar ice loaded under biaxial stresses is indeed closed towards large biaxial stresses (see e.g. [3]). However, the shape of the closure is significantly different from the one proposed on Fig 2 of this manuscript, and failure under such high confinement occurs through spalling, a failure mechanism that is not observed, to my knowledge, in the field (although out-of-plane failure mechanisms might be related). In addition, internal sea ice stresses recorded in the field never reach such strongly confined biaxial stresses, see e.g. [4]. Therefore, a second question arises:
-  What can be effect of such closure on the model outputs? I would suggest the authors to compare simulations performed with and without this closure to analyze this point. If the effect is limited, as I suspect from above, then the introduction of such weakly-justified closure would represent an unnecessary complication. If some impact is observed on the formation of ice arches and/or ridges, such sensitivity analysis would be useful to understand its origin.

9) L180-184, as well as L393. About "the lack of strain hardening in the MEB model leads to non-physical results in convergence with the absence of ridge propagation in the direction parallel to the second principal strain (maximum axial compressive strain)."
Here, the authors reference (Richter-Menge et al., 2002) on the subject of strain-hardening observed in sea ice. (Richter-Menge et al., 2002) themselves refer to the parameterization of strain-hardening of Hibler (1979), where the maximum compressive strength of the ice is proportional to its thickness P = P*h *exp(-C(1-A)). The *same proportionality is actually used in the MEB* (and EB) rheology. Indeed, instead of writing $E \times d, \eta \times d, \sigma_c \times d$ for the strength parameters in the constitutive equation, and writing the constitutive and momentum equations in terms of a vertically integrated stress, *Dansereau et al., 2017, Bouillon et al., 2015, Rampal et al., 2016, Rampal et al., 2019, etc.* all used *stress, instead of the vertically integrated stress,*

and write the rheology term in the constitutive equation as $\nabla \cdot (h\sigma)$. This discussion about "strain-hardening" should be reconsidered in light of this.

In addition, strain-hardening as the result of damage is not supported by experiments on brittle or quasi-brittle materials. A classical illustration is known as the Kaiser stress-memory effect: If a material is damaged up to a given stress, unloaded, and then reloaded, damage will start again when the previous stress will be reached again (e.g., *Heap* 2009).

In case of sea ice, and particularly in the context of ice/structures interactions, the strengthening of crushed, *and then recrystallized*, ice has been discussed in the literature (e.g. [5]). This process however involves various mechanisms such as sintering of crushed grains, refreezing, which are not only mechanically driven. Consequently, a change in critical stress when the material fail remains to be observed, proven or disproven in the case of sea ice, at the geophysical scale, before formulating physical parameterizations for it.

9) Section 2.2.3, and L224-225 "Note that if $\lambda_0$ is sufficiently high, the MEB rheology reduces to the Elasto-Brittle rheology (Bouillon and Rampal, 2015; Rampal et al., 2015)." For the EB rheology, cite [6] rightly instead.

10) Section 3.2.2, and L534-539. About the flow rule and Mohr-Coulomb *failure* criterion:
"In the MEB model, the angle of fracture does not follow the theory. We speculate that the deviations are related to the absence of a flow rule linking the deformations to the yield curve and the angle of internal friction." This is confusing. Fracture occurs in an undamaged or partially damaged material. The material "flows", or undergoes large deformation, once fractured. Therefore, why is the *flow* law determining the angle of the fractures that precedes the flow? Please explain the mechanism behind this.

Second, please note that a flow rule is not required to close the system of equations in the case of the MEB (viscous-elastic-brittle) model. Note also that the statements from lines 79-81 and 192 are contradictory ("We also show that the simple stress correction used in the damage parameterization corresponds to a flow rule" "This correction does not correspond to a flow rule"). Note also that no flow rule has been determined for sea ice from in-situ observations, while the normal flow rule is not supported by lab-scale observations [4].

"In theory, the angle of internal friction governs the intersection angle between lines of fracture (Marko and Thomson, 1977; Pritchard, 1988; Wang, 2007; Ringeisen et al., 2019)":
Recent and extensive work on the observation and modelling of the failure and localisation of deformation in brittle and granular materials (not just sea ice) have demonstrated that the relationship between the angle of internal friction and the intersection angle between conjugate faults is actually more subtle than predicted by the Anderson's theory of faulting: e.g., [7-12].
Initially, the Mohr-Coulomb criterion was not implemented in the MEB rheology (and similarly, the internal friction angle not tuned) in order to fit observations of conjugate faults angles in sea ice. It was rather chosen on the basis of stress measurements within sea ice (see [4]) that suggest a reasonably good fit to this criterion (see [2] on that point).
It has been recently shown that, for an elasto-brittle damageable solid, the fault orientation is not given by the Mohr-Coulomb criterion and Anderson's hypothesis, instead depends on various factors such as the nature of disorder, the Poisson's ratio, or the confinement [12]. It might be interesting in the future to better constrain the MEB parameterization on this basis, comparing simulation results with large-scale observations of leads/faults within the sea ice cover.

References:

[1]    V. Dansereau, J. Weiss, P. Saramito, P. Lattes, and E. Coche, The Cryosphere **11**, 2033 (2017).

[2]    V. Dansereau, J. Weiss, P. Saramito, and P. Lattes, The Cryosphere **10**, 1339 (2016).

[3]    J. Weiss and E. M. Schulson, J. Phys. D: Appl. Phys. **42**, 214017 (2009).

[4]    J. Weiss, E. M. Schulson, and H. L. Stern, Earth Planet. Sci. Lett. **255**, 1 (2007).

[5]    I. J. Jordaan, Engineering Fracture Mechanics **68**, 1923 (2001).

[6]    L. Girard, S. Bouillon, J. Weiss, D. Amitrano, T. Fichefet, and V. Legat, Annals Glaciol. **52**, 123 (2011).

[7]    J.-P. Bardet, Computers and geotechnics **10**, 163 (1990).

[8]    B. Haimson and J. W. Rudnicki, J. Struct. Geol. **32**, 1701 (2010).

[9]    A. Haied, D. Kondo, and J. P. Henry, Mechanics of Cohesive-frictional Materials: An International Journal on Experiments, Modelling and Computation of Materials and Structures **5**, 239 (2000).

[10]    A. Hackston and E. Rutter, Solid Earth **7**, 493 (2016).

[11]    K. Karimi and J.-L. Barrat, Scientific reports **8**, 4021 (2018).

[12]    V. Dansereau, V. Démery, E. Berthier, J. Weiss, and L. Ponson, Physical review letters **122**, 085501 (2019).

This review has been prepared with the collaboration of V. Dansereau

---

## Referee Comment (RC3) · Anonymous Referee #3 · 23 Dec 2019

Review of "The material properties of ice bridges in the Maxwell Elasto-Brittle rheology" by M. Plante, B. Tremblay, M. Losch, and J-F. Lemieux.

The manuscript introduces an implementation of the MEB rheology in the McGill sea-ice model and outlines an idealised test-case studied with the model. The paper discusses the experiment results in relation to expected results from theoretical physical grounds, as well as outlining a few sensitivity experiments done on key parameters. The paper is generally well written and understandable. The science is reasonably interesting and good enough to warrant publication in The Cryosphere. I must say though that the paper quite esoteric, caters to a very narrow audience, and has relatively weak conclusions. As with all idealised, large-scale sea-ice experiments, this one suffers from the fact that comparison to theory, as well as the generalisation of the results, is

very difficult.

It is interesting to see a new implementation of the MEB rheology, which as far as I know has so far only been implemented by Danserau et al (2016) and Rampal et al (2019). Also, even though the setup is virtually the same as that of Dansereau et al (2017), the authors of this paper still to point out some interesting characteristics of the MEB rheology, as their approach is sufficiently different from that of Dansereau et al (2017). The main weakness of the paper is that even though there are some interesting points made (e.g. about the lack of strain hardening and the presence of numerical errors), then those are largely lost to other less interesting aspects (e.g. attempts to estimate physical parameters which should be estimated from a realistic setup). Ideally, the authors should reassess what is really interesting and novel here and focus on those aspects.

Other general comments:

\*) The abstract should be rewritten, as it does not fit well enough the contents of the paper itself.

\*) The description of the MEB model is much too detailed. It should suffice to briefly describe those parts of the model that are particularly relevant to the experiments conducted here, as well as those points where the current implementation differs from that of Dansereau et al and Rampal et al. The differences should also be justified.

\*) The discussion of the cohesion (3.2.1) should take the following into account: Cohesion scales with the model resolution, so you cannot recommend one cohesion value for all resolutions (Weiss et al., 2007, Schulson et al 2009, Rampal et al 2016) Comparing ice bridges across different straits should take the ice thickness into account. Ice bridges longer than 100 km were probably a regular feature of the Kara Sea fast-ice cover (Divine et al., 2005, Olason, 2016) - although this is changing with a thinning ice cover there.

*) Angle of internal friction (3.2.2): I'm not convinced this is an appropriate setup to discuss the internal angle of friction. I would at least have wanted to see variations in the domain geometry, or better yet a model run with the setup from Ringeisen et al (2019).

*) Conclusions: You have a tendency to restate speculations from the text as demonstrable conclusions in the conclusions section. This is a serious fault which cannot be allowed to stand.

Specific comments:

L45: "minimum viscosity" should be "maximum viscosity"

L72: The term "brittle" refers to a certain type of plasticity, so you cannot contrast brittle and plastic (as in "i.e. Brittle [sic] in the MEB, plastic in the EVP"). Sea ice is a brittle plastic, but it can be argued that the (E)VP gives a (too) ductile behaviour to accurately represent sea ice.

L75: It should be "an MEB rheology", not "a MEB rheology"

L75: "implemented in an Eulerian finite difference VP model" - you should elaborate to make this clearer. I didn't understand what you meant before reading your section 2.3.

L122: Strike ", or creep," as creep usually refers to very slow viscous deformation of the ice (ductile deformation), but the viscous part of the MEB represents the stress relaxation that occurs after a brittle rupture.

L125: Rewrite the sentence "This brittle component . . ." emphasising that both models are plastic, but MEB is brittle while VP is ductile.

Section 2.2.1 seems unnecessary (or at least needlessly long) as it's a repetition of previous work.

Ditto for section 2.2.2, except for the point about the lack of strain hardening, a point I don't recall being discussed before in the literature. The authors would do well to

develop this point further and highlight it in their experiments.

L207: Replace lowercase \phi with uppercase \Phi (as well as throughout the rest of the text I believe)

L223: Missing unit for \lambda_0

L224: I don't think it's true that for high enough \lambda_0 MEB becomes EB. At any rate, Bouillon and Rampal (2015) and Rampal et al., (2016) are the wrong references for such a statement.

Section 2.3: After 5 pages of model description we (finally) have something novel. I dare say only the most attentive reader will make it this far, which would be a pity. You should highlight sections 2.3.1 and 2.3.3 and severely shorten everything else in section 2.

L293: Both Rampal et al. (2016) and Dansereau et al. (2016) use the finite element method for the spatial discretisation. Rampal et al., however, use a Lagrangian advection scheme. Section 2.3.3: How does your approach differ from the fixed point iteration used by Dansereau et al. (2016)? As always for numerics the practical implications of performance and accuracy are paramount.

Section 3: The figures should appear in the order they are referred to in the text.

L356: "This deviation results from the absence of a flow rule in the MEB model" This is a very strong statement, but you never sufficiently show this to be the case.

L369: The statement "[n]ote that unless . . . critical stress" is true, and a key aspect of the MEB model as fracturing increases the damage but does not influence the critical stress. Changing the critical stress would be a completely different approach. You need to justify the "in contrast to real ice features" much, much better for that statement to stand.

L378: I find the use of the word "point" in relation to the figures confusing. Can you use

"panel" instead?

L397: The sentence "A physical solution ..." cannot be allowed to stand as it is. It implies that the approach of Rampal et al. is unphysical, without stating why this is so. It also implies that the suggested approach is physical, but the support given is meagre in terms of physics. What is more, I see no physical reason to relate the yield curve parameters to ice thickness.

L402: I found this discussion interesting, but it's tagged onto a very descriptive part of the paper and unlikely to receive much attention as it stands.

L536: I would not describe sea ice as being granular here. It can be, but the central pack, which MEB should describe, is not - nor is the unbroken ice cover the fractures are propagating through.

L535: Here you state that the discrepancies between simulated and expected fracture angles are due to the use of a scalar damage parameter. However, in the text itself, you appropriately say that you _speculate_ that this is the case. You should also use this formulation in the conclusions, as you never conclusively show why you don't get the fracture angles you expect.

L544: You never showed that these errors are not detectable in a different configuration.

L547: You don't show that the use of a damage tensor and a different stress correction scheme would solve the problem.

L558: Recommendations for who? You've only shown idealised experiments, so it is very hard to recommend anything to people wanting to run a realistic setup.

L559: Again, in the idealised setup you need this - but what is the impact in other scenarios? You should at least make that distinction clear.

L562: You never show this to be the case, it, therefore, doesn't belong to the conclusions, and certainly not to your recommendations.

L564: Ditto for this point; you never show this to be the case, it, therefore, doesn't belong to the conclusions, and certainly not to your recommendations.

References: I dislike typesetting dois as URLs or the inclusion of both a doi and URL for a single reference. There is also a discrepancy in the capitalisation of the paper titles.

L674: Remove an extraneous ˆ

L735: The full version (not the "discussions" one) is 2016.

Figures:

Figure 5: The phrase "Numbers indicate the location . . ." is not descriptive. Please rewrite.

Figure 8: The last word of the caption is misspelt ("componant", instead of "component").

Figure 10: Please write what the arrows on the yield curve plot indicate.

---

## Author Comment (AC1) · 16 Feb 2020

**Answers to tc-2019-210 RC1**

**February 15th, 2020**

Note :

- The referee comments are shown in black,
- The authors answers are shown in blue,
- *Quoted texts from the revised manuscript are shown in italic and in dark blue.*

The manuscript named "The material properties of ice bridges in the Maxwell Elasto-Brittle rheology" test the MEB sea-ice rheology in a traditional finite difference framework. The aim is to investigate the damage parameterisation. This is achieved with an idealized model setup of a channel that is narrow in the middle and wider in the two ends. I think that this is a very relevant to study the damage parameterisation as this (at least in my opinion) is important for new developments within sea-ice dynamics. The manuscript is in general well written and therefore fairly easy to read.

I could wish for a better organization of the figures. In addition, the result section would be much easier to read if the figures were numbered in the order they are referenced. At last I think that the simplifications made in terms of zero ocean currents/sea surface tilt may have bigger impact, especially in some of the examples mentioned within the Canadian archipelago, where tides are significant. This is not necessary to include within this study but a better discussion of the limitation would be nice.

The results in general seems less impressive than other studies using the MEB rheology. A discussion of the performance of these would also be nice. Many of these points are mentioned but I think that it would be beneficial to collect these in a discussion and maybe spend a few more words.

We thank the referee for his or her thorough review of the manuscript and constructive comments.

**Major revisions:**

I think that the focus of the abstract is a bit off and I would like this to be revised. It is not that important that the framework of the sea-ice model originally was build for VP dynamics as this is not mentioned in the manuscript. The eulerian/lagrangian implementation is more relevant. I would like the abstract to include line 77-82 as these
fit well into a summary/abstract and less into the introduction. Corrected as suggested by the reviewer

>> As suggested by the reviewer, the abstract was re-written to better reflect the manuscript's content.

We have kept however the sentence about the implementation of the MEB model on the Eulerian, finite differentiation framework, because this allows for a direct comparison with other models commonly used in GCMs. This is not a trivial task and many have tried without success. This is considered a significant contribution for the ice modeling community that is worth reporting in the abstract. Instead, we have put more emphasis on this in the body of the paper to justify its inclusion in the abstract.

Discussions are scattered around in the manuscript. I would like a collected discussion.

>> We added a new section where we collected our discussions on the damage parameterization, its instabilities and the orientation of the lines of fractures.

–   What improvements/limitations are there when the framework moves from a Lagrangian to an Eulerian approach.

>> The limitations of moving from a Lagrangian to a Eulerian framework is mainly linked with the advection of tracers within the model. A Lagrangian approach allows to follow deforming elements within the domain resulting in low numerical diffusion. The disadvantage of this approach is that one must recalculate a new grid periodically (called regridding) when the elements are too distorted, requiring the use of interpolation techniques that leads to diffusion of resolved features and requiring a significant amount of CPU time. In a Eulerian framework, higher order advection schemes are typically used (but not here) to limit numerical diffusion and the model grid remains the same throughout the full integration, resulting in lower CPU cost. Note also that the use of a Eulerian scheme is not novel here: the MEB model was originally implemented in a Eulerian scheme in Dansereau et al., 2016, although using Finite Element Methods.

The goal here is not so much about model improvements (or limitations), but rather about advantages of having two different models on the same platform. This has been clarified at L221-226 in the revised manuscript. Namely, the advantages of coding the MEB model using a Eulerian, finite differentiation framework is that it allows for a direct comparison of the MEB model physics (rheology, yield curve, deformations) with that of the standard viscous plastic approach (or variation thereof) used in the vast majority of GCMs and coupled ice-ocean models, independently of the differences in the numerical framework (i.e. Lagrangian vs Eulerian advection scheme, regridding, the use of Finite Element Methods and triangular mesh).  The caveats of using the Finite Difference scheme is also discussed in section 2.3.2, at L267-283.

The study limits the effects of the ocean (eg. tides) by neglecting it. For an idealized study like this it is fine just to look at the wind. But in terms of comparisons with real data then this restricts the value og the study. Tides are very high from Kanes Basin and southwards. This is an important factor when the stability of the last fast ice is to be considered and compared with real life. This is briefly mentioned on line 500, however I would like a bigger discussion of this.

>> We agree with the reviewer that the tides and thermal stresses are important in the landfast ice break up process. Here, we consider our "wind forcing" to be representative of the combined surface forces on the ice from both winds and ocean currents including tides, which vary depending on the location. This was clarified in the revised manuscript by re-writing the surface forcing into a landfast ice forcing term independent of the ice velocity and a water drag term that is only significant in drifting ice. As such, the forcing imposed on the ice bridge is no longer assumed to originate from the winds. The simulations are now discussed in terms of forcing values, rather than wind magnitude. Note that our ideal simulations and cohesion estimates are not sensitive to the source of the forcing, only to its

magnitude and direction. This is now clarified at L85-L96. The wind forcing and surface current values associated to the forcing used to derive the material properties of sea ice is also indicated at L410-411. We believe that a lengthy discussion on these factors is outside of the scope of this paper, which is concerned with the simulation of ice arches in the MEB framework and the influence of the material properties in their formation and stability. We also clarified the scope of the study in the introduction so as to not create expectations that are not met in the body of the paper.

Figures are very inconsistent when labeling. These should be changed. I have suggested updates to almost all of the figures. These comments are in the minor correction/technical correction part.

>> There was a duplicated figure in the manuscript that led to confusion in the automatic latex-referencing. This addressed several of the comments raised by the reviewer. We apologize for not noticing this before submission. We have corrected all other issues as proposed by the reviewer.

**Minor details**

Line 2- Please revise sentence. An example is provided: The effect of the material parameters on ice arches in a numerical framework that includes both the Maxwell Elasto-Brittle (MEB) including a damage parameterization and the Viscous-Plastic (VP) dynamics.

>> The abstract was completely rewritten.

Following lines after line 2: I assume that this is MEB but it is a little unclear

>> The abstract was rewritten.

Line 20 I think that this should be reformulated. For instance, ice keel don't protect sea-ice from forcing. It creates a friction that resist the forcing. I would reformulate this

>> This was reformulated as suggested by the reviewer, and now reads:

*Typically, large landfast ice areas can form and remain stable due to the presence of islands or by the grounding of ice keels on the ocean floor.*

Line 35 Ice thickness anomaly is this in time or space? I guess that the influencing factor is the current ice thickness, spatial variation (anomaly).

>> This was referring to ice thickness anomalies from year to year. This is clarified in the revised manuscript and now reads:

*A variety of studies suggest that the inter-annual variability in presence or absence of ice arches in given locations are influenced by several factors, such as ice thickness anomalies [...]"*

Line 47: replace new rheology with new rheologies

>> Corrected as suggested by the reviewer.

Line 67. References to figures in other articles makes it hard to read. Please either add the location on figure 1 or add a map where this can be shown.

>> The locations were added in Figure 1, as suggested by the reviewer.

Line 77 to 82: This part would be well suited for conclusions and/or abstract. The introduction should be more overview of previous studies and an overview of what will be presented. Not results.

>> It is correct that such statements are usually only included in the abstract or conclusions. However, including those at the end of the introduction situates the reader up front and allow him/her to focus on "how we arrived at those conclusions". It is a style adopted by many authors both in oral presentations and written papers, and recommended in the book by Joshua Schimel "how to write papers that get cited and proposals that gets funded". We believe it leads to a more active form of presentation that is more engaging for the reader. For this reason, we have opted to leave it in the introduction.

Line 95: Nares Strait has strong tides in the part near Baffin Bay, thus the ocean currents would most likely be an important factor especially before and probably after the fast ice region has formed. Therefore, this should be mentioned in the discussion.

>> As we state in response to the general comments above, the simulations are not sensitive to the source of the forcing, only to the total magnitude. We also rephrase the results so that we refer to the forcing magnitude, rather to the wind speed throughout the revised manuscript. Note also that the regions of high tidal forcing downstream of Nares Strait is rarely landfast (see Hannah et al., 2009, Vincent 2019). This is also clarified at L413-414:

*"Note that higher forcing may be frequent in areas associated with strong tides, although these locations correspond to unstable landfast ice areas and recurrent polynyas (Hannah et al., 2009)."*

Coriolis is only zero when the ice is not moving.

>> We are mainly concerned with the loading of landfast ice until the break-up and in the derivation of constraints on the mechanical properties of landfast sea ice. It is correct that the subsequent motion after break-up will have small errors (of the order of 10%, Turnbull et al. 2017) given that ice is relatively thin and that the Coriolis term scales with ice thickness. This is clarified on L80-81 of the revised manuscript:

*"We assume the ocean to be zero and ignore the Coriolis term, as it is identically zero for immobile ice. These assumptions are appropriate for landfast ice, but could result in small errors in drifting ice (Turnbull et al., 2017)."*

The discussion of the influence of the ocean is too small.

>> See comment above. The analysis was re-written such that the forcing used on the ice bridge is not exclusively coming from the atmosphere, but can also originate from the ocean. This is clarified at L85-96 in the revised manuscript.

Equation 6 ":" in the equation?. This is described in equation 8. This should be moved here (first place that it is used)

>> Corrected as suggested by the reviewer.

Line 119 lhs and rhs should be written without using a abbreviation.
>> Corrected as suggested by the reviewer.

Line 120- 124. These sentences are a bit hard to follow. Please revise.
>> These sentences were clarified, as suggested by the reviewer. They now read:

*Eq. 6 indicates that in the visco-elastic regime (before fracture), the deformations are dominated by a fast and reversible elastic response (first term on the left hand side of Eq. 6), with a slow viscous dissipation acting over longer timescales (second term on the left hand side). The reversibility of the elastic deformations implies that the elastic strains return to zero if all loads are removed. This results from a memory of the previous elastic stress and strain states given by the time-derivative in Eq. 6. The Maxwell viscosity term, although orders of magnitude lower that the other terms in the visco-elastic regime, leads to a slow viscous dissipation of this elastic stress memory over long timescales determined by λ.*

Line 265 Figure 3 No need to show a ARAKAWA grid. This is a standard. I would remove the figure.

>> This figure is removed, as suggested by the reviewer.

Line 317 Remove "-" 2 times

>> Corrected as suggested by the reviewer.

Line 404 In short the physical solution did not converge until the tolerance is lower than 10^-10. How many iterations are required? Is this important for the computational time (how important?).

>> The errors are not due to a difficulty in solving the equations, but rather to the fact that the residual errors are accumulating in the memory terms (instability). The MEB rheology actually converges rapidly, especially given the small time step required by the CFL criterion. The convergence is most time reached within 6-8 outer-loop iterations (fgmres converges in only 1 iteration, given the very small changes in the solution in 0.5s.). This is clarified at L496-500 in the revised manuscript.

Also, using a very low residual tolerance does not solve the problem. This is now better illustrated in Fig. 15 in the revised manuscript. Note that the small timestep, however, is a burden in terms of total time of integration, especially if longer-term simulations are needed. Using a low tolerance increases that burden. For example, on a standard computer (Quad Core Intel Xeon E5-1630 v3, L2 cache of 10.0 MiB with a RAM of 62.63 GiB), the 10h simulations are completed in 4h30 when using a tolerance of 10^-10, 2h30 when using a tolerance of 10^-4, and 1h30 when using the VP model and a time step of 10min, a tolerance of 10^-3 and a maximum of 500 outer loop iterations.

Line 447 Nature is a bit more complex than just wind. Orography ocean currents etc. also play a role, thus values like the cohesion of sea ice should be smaller than 21KN/m seems to be a very rough estimate based on parts of the momentum equation. Admitted wind is normally the dominant factor along with the resistance (internal
strength)

>> As stated in general comments above, we agree with the reviewer that the tides and thermal stresses are important in the landfast ice break up process. In our ideal simulations, we consider our forcing to

be representative of the combined surface forces on the ice, which vary depending on the location. This was clarified at L85-96 and throughout the analysis. The forcing used to derive our cohesion estimates (0.15 N/m$^2$) is consistent with a typical forcing on landfast ice (10m/s winds or 0.15m/s current). This is now clarified at L407-408. We also changed the wording throughout the text to discuss the simulation in terms of surface stresses rather than wind magnitude.

Line 473 How does this compare with results from other MEB implementations.

>> To our knowledge, there are only 2 other implementations of the MEB model: The model of Dansereau et al., (2016, 2017) and NeXtSIM (Rampal et al., 2016, 2019). The angles of fracture and type of deformation associated with the damage have not been investigated in details in those studies, but was recently investigated in Dansereau et al. 2019, who demonstrated that the orientation of the lines of fracture does not follow those predicted by the Mohr Coulomb theory. Our findings are consistent to this assessment. These clarifications are included in section 4 in the revised manuscript.

Line 520: is the ":" suppose to be there?

>> Removed, as suggested.

Line 528: It would be very interesting to include this in a VP/EVP model.

>> We agree. This is something that we are currently working on.

Figure 5 text. Top panel? I can only see one panel in figure 5.

>> Corrected as suggested by the reviewer.

Figure 6. This figure should be labeled a through d instead of a1 2 3 b

>> The numbers are referring to the points in Fig 5, which correspond to the damage fields. The figure label was clarified.

Figure 7 Same as 7, Which points?

>> This error in the label is corrected as suggested by the reviewer.

Figure 8: I would say colored dashed lines

>> This error in the label is corrected as suggested by the reviewer.

Figure 10 Dots are very hard to see. It would be nice to increase the size of these.

>> This figure was removed.

Figure 11: Which colored lines? They are defined in figure 8. Are they the same?

>> Removed, as suggested by the reviewer. There are no colored lines.

Figure 12: I would still label these a, b,c and d. Then add to the text.

>> This figure was removed.

Figure 13 which colored lines? Are they the same?

>> This label error is corrected as suggested by the reviewer.

Figure 14. Please use a,b,c: : : References to the residual tolerance are not very easy. Left 10ˆ-6 and right 10ˆ-10 does not make sense.

>> Corrected, as suggested by the reviewer.

Figure 17 Arrows are very hard to see.

>> Corrected, as suggested by the reviewer.

Figures in general should be in order of them being mentioned in the text. For instance the result section seem to jump back and forth. I assume that when done with the review process they need to be inserted at appropriated places.

>> As state above, there was a duplicate figure in the manuscript. Removing this figure has resolved this issue. We also removed Figure 10 from the submitted manuscript, which ease the figure flow.

Dansereau, V., Weiss, J., Saramito, P., Lattes, P., and Coche, E.: A Maxwell-Elasto-Brittle rheology for sea ice modeling, Mercator Ocean Quarterly Newsletter, pp. 35–40, 2015.

Dansereau, V., Weiss, J., Saramito, P., and Lattes, P.: A Maxwell elasto-brittle rheology for sea ice modelling, The Cryosphere, 10, 1339–1359, https://doi.org/10.5194/tc-10-1339-2016, 2016.

Hannah, C.G., Dupont, F., Dunphy, M., 2008b. Polynyas and tidal currents in the Canadian arctic archipelago. Arctic 62, 83–95

Rampal, P., Bouillon, S., Ólason, E., and Morlighem, M.: neXtSIM: a new Lagrangian sea ice model, The Cryosphere Discussions, 9, 735 5885–5941, https://doi.org/10.5194/tcd-9-5885-2015, http://www.the-cryosphere-discuss.net/9/5885/2015/, 2015.

Rampal, P., Dansereau, V., Olason, E., Bouillon, S., Williams, T., and Samaké, A.: On the multi-fractal scaling properties of sea ice deformation, The Cryosphere Discussions, 2019, 1–45, https://doi.org/10.5194/tc-2018-290, https://www.the-cryosphere-discuss.net/tc-2018-290/, 2019.

Turnbull, I. D., Torbati, R. Z., and Taylor, R. S. ( 2017), Relative influences of the metocean forcings on the drifting ice pack and estimation of internal ice stress gradients in the Labrador Sea, J. Geophys. Res. Oceans, 122, 5970– 5997, doi:10.1002/2017JC012805.

Vincent, R.F. A Study of the North Water Polynya Ice Arch using Four Decades of Satellite Data. Sci Rep 9, 20278 (2019). https://doi.org/10.1038/s41598-019-56780-6

---

## Author Comment (AC2) · 16 Feb 2020

**Answers to tc-2019-210 RC2**

**February 15th, 2020**

Note :

- The referee comments are shown in black,
- The authors answers are shown in blue,
- *Quoted texts from the revised manuscript are shown in italic and in dark blue.*

This manuscript presents an implementation of the recently developed Maxwell Elasto-Brittle rheology for sea ice mechanics within a finite-difference scheme, and the realization of idealized simulations of ice deformation and motion within a channel. This work is quite similar with what was done in [1], however with several differences:

- A finite-difference, instead of a finite-element, numerical scheme is used - the numerical implementation of the MEB rheology is performed exploiting the code framework of a standard VP code.
- the initial condition is a purely homogeneous ice plate (constant thickness, constant elastic and strength properties), without any sort of initial disorder implemented

The effects of material properties and model parameters are then analyzed from a sensitivity analysis using strictly symmetric boundary conditions and geometry.

Overall, this manuscript is clearly written. Most of the results are consistent with [1], proving the robustness of the MEB rheology to adequately simulate sea ice damage, fracture, or strain localization. I am therefore rather favorable to a publication in The Cryosphere. There are however several mistakes, misunderstandings, or points to discuss more thoroughly (e.g., the flow rule and its relationship to the angle of fractures), which should be fixed before final acceptance. They are listed below:

We thank the referees, Dr. Weiss and Dr. Dansereau, for their thorough review of the manuscript and constructive comments.

1) Title: "The material properties of ice bridges…". This is strange, and unclear: bridges do not have "material properties". Maybe the authors wanted do say: "The effect of material properties on the simulation of ice bridges in the .." ?

>> We agree. The title was rephrased as:

*"Landfast sea-ice material properties derived from ice bridge simulations using the Maxwell Elasto-Brittle rheology"*

2) p3, L65-73, as well as p14, L390-395: The authors argue at different places about the position of the ice bridges, up- or down-wind the channels. This argumentation is not very clear:

– to what extent the situation shown on fig. 1b is systematic ? To the best of my knowledge, ice bridges can take place at different positions along channels (as actually shown on fig. 1b), including along Nares strait (see e.g. fig. 1 of [1]), and I do not know a systematic, statistical analysis of that (but would be happy to learn about such analysis, if any).

>> On L46-53 (in the revised manuscript), we refer to the fact that ice bridges simulated by the MEB model are mostly located downstream of narrow channels, rather than upstream (i.e. as seen in Figs 4,5,7 and 9 in Dansereau et al. 2017). We do not refer to a single specific location. In the Nares strait, the (stable) ice arch does have preferred locations. Fig. 1 from Dansereau et al., 2017 shows some of these positions, usually either in the Lincoln sea (Fig. 1a and b) or in Kane basin (Fig. 1c). These positions can be seen in several landfast ice cover assessments (see for instance Tivy et al 2011, Galley et al. 2012, Yu et al 2013), and are also recently fully described in Vincent (2019).

– in [1], ice breakup occurs in several successive steps, at different locations along the Nares strait. In conclusion, I am not sure that the unique observation of fig 1b can serve as a severe constrain on model parameterizations.

>> We agree with this statement. The break-up of landfast ice is a rapid process during which the temporary ice edges are highly dependent on the pre-existing ice deformations. We cannot reproduce this in our idealized simulation.  Here, we rather refer to the formation of stable ice bridges, which are less influenced by these factors. Figure 1 shows such an ice arch in Kane Basin, which remained in place from early March 2018 to mid-June 2018. We have clarified this in the revised manuscript at L46-53, L470-476 and in the figure caption. We also specify that the aim of the paper is the necessary conditions for the simulation of stable ice arches rather than the temporary ice arches formed in the process of landfast ice break-up. In Dansereau et al. 2017, the location of the stable ice arch (i.e. in Figs 4,5,7 and 9) are located downstream of the narrow channels, seldom seen in the Nares Strait (see Vincent 2019).

3) L92-93. Indeed, at the timescales involved in classical climate models (time step of several hours, as in the papers cited), the advection term can be neglected. This might not be true when considering a much smaller model time step (~s). What is the time step of the present simulations? Have you done a proper scale analysis taking that time step into account?

>> The time step used in all simulations is 0.5s (now specified at L189 of the revised manuscript and corrected in Table 2). In such a small time scale, the advection term is many orders of magnitude lower than the inertial term (the inertial term scales as $1/T$, T being the time scale). The advection term can however become important at small length scales. In drift ice, it scales as $\rho_i h_i U^2/L \quad 10^{-3}$ N/m$^2$ , where $\rho_i$ (~900kg/m$^3$) is the ice density, $h_i$ (~1m) is the ice thickness, L (~2-10 km) is the space resolution, U ~0.1m/s is a typical ice velocity. This is three orders of magnitude smaller than a characteristic surface wind or ocean stress. At the edge of an ice arch, where a discontinuity in sea ice drift is present at small scales (2 km in our case), it remains two orders of magnitude smaller than other terms in the momentum equation. This has been clarified in L80-85 of the revised manuscript.

4) In section 2.1, ice thickening through mechanical redistribution when A=1 is not considered. Was such redistribution scheme implemented? If not, this would be a problem, as such scheme, even very simple, and coupled to the MEB rheology, was found to generate realistic ITD [1]. This would therefore likely affect all the discussion about ice ridges throughout the entire manuscript. If such scheme was implemented, please detail.

>> Mechanical redistribution is taken into account in our simple 1-category model (i.e. ice or open water). When A=1 and sea ice convergence occurs, the mean ice thickness increase (see continuity Eq. 4 in the manuscript), but since A=1 is capped at one, this leads to the actual thickness of ice in a grid cell (h/A) to increase, i.e. ridging. A simple 1-category model does not resolve the ITD per se, unless the variability in ice thickness is resolved (i.e. unless the model is run at O(1m) resolution, at which sea ice no longer behaves as a 2D material.

5) Equation (1): C is not defined here, but much later (eq. (9)). Please modify.

>> Corrected as suggested by reviewer.

6) L125-127: I would not agree and there seems to be a misunderstanding here: the elasto-brittle component of the MEB rheology is, by construction, associated with small (and reversible) deformations, while the Maxwell component deals with large (and irreversible) deformations. This is therefore fundamentally different from the VP model where the plastic regime is associated with irreversible strains but strain-rate independent stresses (while the Maxwell component of the MEB model is indeed strain-rate strengthening and not strain-rate independent). Please justify in physically-sound way or remove that sentence.

>> These lines were re-written to clarify this statement. As the reviewer points out, the small deformations in the EB component are elastic or reversible, and they are not in the VP model. Here we make the observation that during the fracture process, the larger (and partly still elastic) deformations + viscous dissipation associated with the damage are analog to the plastic regime in the VP model. When a fracture is developing, the stress state is kept on the critical yield curve while the strain rates and damage increase, and the Elastic stiffness and viscosity decrease. In the VP model, the non-linear bulk and shear viscous coefficients reduce with increasing strain rates, such that the stress states (the product of the two) remain on the yield curve. Thus, in both models, the stress state is independent of the deformation rates during the fracturing. The two models do differ, as stated by the reviewer, post fracturing as the VP model does not have a memory of past deformations other than via the continuity equation and its impact on the ice thickness and concentration. In the MEB, the post-fracture elastic deformation remains important unless the damage is large (d>0.8), while deformations are viscous in fully developed fractures. The damage corresponds to a material memory of past deformation. The text in section 2.2 was heavily re-written to clarify this physics, at L119-124 and L204-219.

7) L134-137 and L222: At least in the case of (Tabata, 1955) and (Weeks and Assur, 1967), these authors discuss the creep of bulk saline ice, driven by viscoplasticity at the crystal scale. The concept introduced in the MEB rheology is fundamentally different: a linear viscous term is introduced to account for the cataclastic flow of highly damaged ice, and associated stress relaxations.

>> Here we were referring to the dissipative effect of the viscous term in undamaged ice. We agree that this effect is negligible in terms of deformation in the landfast ice (for example, a sustained stress of 50kN/m in our model results in a viscous creep deformation of the order of $10^{-5}$), but is significant in

dissipating the elastic stress memory over a long time scale. This is clarified at L119-124 in the revised manuscript.

8) L153-154: This statement is wrong: plane stresses were considered in Dansereau et al., 2015, 2016, 2017 and any implementation of the MEB rheology. This is indeed the correct assumption for thin plates. Note however that the impact of such assumption (plane stress vs plane strain) has little consequence on the global behavior. Note also that the constitutive equation present in the early newsletter Dansereau et al., 2015 is that of the generic Maxwell model.

>> The paper of Dansereau et al. (2016) have a factor of 1/(1+v)(1-2v) in the stress-strain relationship, indicating that the authors have used the plane-strain assumption. Dansereau et al. (2017) however do use the plane stress assumption, as pointed out by the reviewer. We now cite the reference Dansereau et al., (2016) in the revised manuscript rather then (2015) in the original version. We state that Dansereau et al., (2017) do use the plane stress assumption for completeness.

9) Section 2.2.2. The authors propose to close the damage envelope towards large compressive stresses using eq. (11). This is another difference with the initial MEB model [2]. In principle, I would say "Why not ?". However, several questions arise:
- What is the physical justification of such closure? At the lab scale, the failure envelope of columnar ice loaded under biaxial stresses is indeed closed towards large biaxial stresses (see e.g. [3]). However, the shape of the closure is significantly different from the one proposed on Fig 2 of this manuscript, and failure under such high confinement occurs through spalling, a failure mechanism that is not observed, to my knowledge, in the field (although out-of-plane failure mechanisms might be related). In addition, internal sea ice stresses recorded in the field never reach such strongly confined biaxial stresses, see e.g. [4].

>> See answer below.

Therefore, a second question arises:
- What can be effect of such closure on the model outputs? I would suggest the authors to compare simulations performed with and without this closure to analyze this point. If the effect is limited, as I suspect from above, then the introduction of such weakly-justified closure would represent an unnecessary complication. If some impact is observed on the formation of ice arches and/or ridges, such sensitivity analysis would be useful to understand its origin.

>> Our concern is not the bi-axial compression state (since, as stated by the reviewer, it is rarely observed) but rather the uni-axial compression which can lead to large compressive stresses – i.e. larger than the mechanical strength of sea ice. This is why we argue for the use of a capping in compressive stress: to limit the uniaxial compression. This is specified at L151 in the revised manuscript. This has the side benefit of improving the numerical stability of the model as discussed in section 4.2 in the revised manuscript. The compression capping does influence the simulation results, as discussed in details in section 3.2.4 of the original manuscript (3.1.2 in the revised manuscript): it can cause uni-axial failure along the upstream coastlines, instead of lines of shear fracture propagating at an angle from the island corners, as in the control run simulation.

10) L180-184, as well as L393. About "the lack of strain hardening in the MEB model leads to non-physical results in convergence with the absence of ridge propagation in the direction parallel to the second principal strain (maximum axial compressive strain)." Here, the authors reference (Richter-Menge et al., 2002) on the subject of strain-hardening observed in sea ice. (Richter-Menge et al., 2002)

themselves refer to the parameterization of strain-hardening of Hibler (1979), where the maximum compressive strength of the ice is proportional to its thickness P = P*h *exp(-C(1-A)). The same proportionality is actually used in the MEB (and EB) rheology. Indeed, instead of writing E x d, η x d, $\sigma_c$ x d, for the strength parameters in the constitutive equation, and writing the constitutive and momentum equations in terms of a vertically integrated stress, Dansereau et al., 2017, Bouillon et al., 2015, Rampal et al., 2016, Rampal et al., 2019, etc. all used stress, instead of the vertically integrated stress, and write the rheology term in the constitutive equation as div(hσ). This discussion about "strain-hardening" should be reconsidered in light of this.

>> This is a good point. The lack of strain hardening in our model is related to the fact that we used the vertically integrated stress definition (div(sigma) and not div(h*sigma) as in other MEB implementations), so that we keep the same numerical/model platform as our standard VP model. It the original submission, we did not adapt the yield criterion accordingly. We now include the thickness dependency in the cohesion (and compressive strength): i.e. c = c0*h *exp(-C(1-A)). This is needed for the set of equation (momentum, stress-strain relation and yield criterion) to be equivalent to the previous MEB model implementations. These clarifications are now added in the model description, in section 2. As expected, using the vertically integrated material parameters does not change the model behavior except for the strain hardening associated to increasing thickness now occurring upstream of the channel, and for a reduced stability of the model (the higher cohesion cause higher compressive stresses and increase the instability issue discussed in the paper). We modified the discussion accordingly, and the comment on strain hardening is removed. We also specify that longer time integration is required for the formation of an ice arch upstream of the channel.

In addition, strain-hardening as the result of damage is not supported by experiments on brittle or quasi-brittle materials. A classical illustration is known as the Kaiser stress-memory effect: If a material is damaged up to a given stress, unloaded, and then reloaded, damage will start again when the previous stress will be reached again (e.g., Heap 2009). In case of sea ice, and particularly in the context of ice/structures interactions, the strengthening of crushed, and then recrystallized, ice has been discussed in the literature (e.g. [5]). This process however involves various mechanisms such as sintering of crushed grains, refreezing, which are not only mechanically driven. Consequently, a change in critical stress when the material fail remains to be observed, proven or disproven in the case of sea ice, at the geophysical scale, before formulating physical parameterizations for it.

>> We agree with the reviewer. The strain hardening in our simulations is because of the h-dependency of the material strength parameters (i.e., a thickening of the ice increases the vertically integrated material strength). It is related to the use of vertically integrated stress, and not to the hardening of the ice material itself. The comment on strain hardening was removed from the analysis.

11) Section 2.2.3, and L224-225 "Note that if λ0 is sufficiently high, the MEB rheology reduces to the Elasto-Brittle rheology (Bouillon and Rampal, 2015; Rampal et al., 2015)." For the EB rheology, cite [6] rightly instead.

>> Corrected as suggested by the reviewer

12) Section 3.2.2, and L534-539. About the flow rule and Mohr-Coulomb failure criterion: "In the MEB model, the angle of fracture does not follow the theory. We speculate that the deviations are related to the absence of a flow rule linking the deformations to the yield curve and the angle of internal friction." This is confusing. Fracture occurs in an undamaged or partially damaged material.

The material "flows", or undergoes large deformation, once fractured. Therefore, why is the flow law determining the angle of the fractures that precedes the flow? Please explain the mechanism behind this.

>> Here, we use the term "fracturing" to represent the development of damage: in the MEB model, the development of the fracture is not instantaneous, and damage increases over several time steps as the deformations progress. As such, the locally increasing deformation influences the surrounding strain orientation. We speculate that this influences the stress concentration associated with the fracture that leads to yielding in neighboring cells (see Dansereau et al.,2019). The ice arch and fracture lines are a result of this propagation of local damage in space. If the orientation of the deformation rate tensor was associated with the yield criterion during this process, we speculate that the lines of fracture would follow the Mohr-Coulomb theory, as observed in other models using a flow rule (see Ringeisen et al., (2019) for instance). In the MEB model, they are not and the fracture line orientation does follow the Mohr-Coulomb theory. This result is consistent with those of Dansereau et al., (2019). We have clarified this in the new discussion section of the revised manuscript, at L477-489.

Second, please note that a flow rule is not required to close the system of equations in the case of the MEB (viscous-elastic-brittle) model.

>> We agree with this comment. The point made here is that the deformations during the development of damage might influence the orientation of the lines of fracture. This is consistent with Dansereau (2019), in which the lines of fractures are found to be determined by the stress concentration and the collective spreading of the damage along lines of damage instability.

Note also that the statements from lines 79-81 and 192 are contradictory (''We also show that the simple stress correction used in the damage parameterization corresponds to a flow rule'' "This correction does not correspond to a flow rule").

>> This was indeed not clear. We removed the 1$^{st}$ sentence, and kept the statement at L171-173 in the revised manuscript, i.e. that the stress correction path does not correspond to a flow-rule.

Note also that no flow rule has been determined for sea ice from in-situ observations, while the normal flow rule is not supported by lab-scale observations [4]. "In theory, the angle of internal friction governs the intersection angle between lines of fracture (Marko and Thomson, 1977; Pritchard, 1988; Wang, 2007; Ringeisen et al., 2019)": Recent and extensive work on the observation and modelling of the failure and localisation of deformation in brittle and granular materials (not just sea ice) have demonstrated that the relationship between the angle of internal friction and the intersection angle between conjugate faults is actually more subtle than predicted by the Anderson's theory of faulting: e.g., [7-12]. Initially, the Mohr-Coulomb criterion was not implemented in the MEB rheology (and similarly, the internal friction angle not tuned) in order to fit observations of conjugate faults angles in sea ice. It was rather chosen on the basis of stress measurements within sea ice (see [4]) that suggest a reasonably good fit to this criterion (see [2] on that point).
It has been recently shown that, for an elasto-brittle damageable solid, the fault orientation is not given by the Mohr-Coulomb criterion and Anderson's hypothesis, instead depends on various factors such as the nature of disorder, the Poisson's ratio, or the confinement [12]. It might be interesting in the future to better constrain the MEB parameterization on this basis, comparing simulation results with large-scale observations of leads/faults within the sea ice cover.

>> Thank you for this comment. We agree that there are many ways other than a normal flow rule to

constrain the orientation of deformations. Here, we point out that the damage parameterization should relax the elastic coefficients in a way that leads to a deformation field that is consistent with observations. This would constitute an improvement to the current damage parameterization.

Dansereau, V., Weiss, J., Saramito, P., Lattes, P., and Coche, E.: A Maxwell-Elasto-Brittle rheology for sea ice modeling, Mercator Ocean Quarterly Newsletter, pp. 35–40, 2015.

Dansereau, V., Weiss, J., Saramito, P., and Lattes, P.: A Maxwell elasto-brittle rheology for sea ice modelling, The Cryosphere, 10, 1339–1359, https://doi.org/10.5194/tc-10-1339-2016, 2016.

Dansereau, V.,Weiss, J., Saramito, P., Lattes, P., and Coche, E.: Ice bridges and ridges in the Maxwell-EB sea ice rheology, The Cryosphere, 11, 2033–2058, 2017.

Galley, R. J., B. G. T. Else, S. E. L. Howell, J. V. Lukovich, and D. G. Barber, 2012: Landfast sea ice conditions in the Canadian Arctic: 1983–2009. Arctic, 65, 133–144.

Rampal, P., Bouillon, S., Ólason, E., and Morlighem, M.: neXtSIM: a new Lagrangian sea ice model, The Cryosphere Discussions, 9, 735 5885–5941, https://doi.org/10.5194/tcd-9-5885-2015, http://www.the-cryosphere-discuss.net/9/5885/2015/, 2015.

Rampal, P., Dansereau, V., Olason, E., Bouillon, S., Williams, T., and Samaké, A.: On the multi-fractal scaling properties of sea ice deformation, The Cryosphere Discussions, 2019, 1–45, https://doi.org/10.5194/tc-2018-290, https://www.the-cryosphere-discuss.net/tc-2018-290/, 2019.

Ringeisen, D., Losch, M., Tremblay, L. B., and Hutter, N.: Simulating intersection angles between conjugate faults in sea ice with different viscous–plastic rheologies, The Cryosphere, 13, 1167–1186, https://doi.org/10.5194/tc-13-1167-2019, https://www.the-cryosphere.net/13/1167/2019/, 2019.

Tivy, A., Howell, S. E. L., Alt, B., McCourt, S., Chagnon, R., Crocker, G., Carrieres, T., and Yackel, J. J. ( 2011), Trends and variability in summer sea ice cover in the Canadian Arctic based on the Canadian Ice Service Digital Archive, 1960–2008 and 1968–2008, J. Geophys. Res., 116, C03007, doi:10.1029/2009JC005855.

Vincent, R.F. A Study of the North Water Polynya Ice Arch using Four Decades of Satellite Data. Sci Rep 9, 20278 (2019). https://doi.org/10.1038/s41598-019-56780-6

Yu, Y., H. Stern, C. Fowler, F. Fetterer, and J. Maslanik, 2014: Interannual Variability of Arctic Landfast Ice between 1976 and 2007. J. Climate, 27, 227–243,https://doi.org/10.1175/JCLI-D-13-00178.1

---

## Author Comment (AC3) · 16 Feb 2020

**Answers to tc-2019-2010 RC3**

February 15ᵗʰ, 2020

Note :

- The referee comments are shown in black,
- The authors answers are shown in blue,
- *Quoted texts from the revised manuscript are shown in italic and in dark blue.*

Review of "The material properties of ice bridges in the Maxwell Elasto-Brittle rheology" by M. Plante, B. Tremblay, M. Losch, and J-F. Lemieux.

The manuscript introduces an implementation of the MEB rheology in the McGill sea ice model and outlines an idealised test-case studied with the model. The paper discusses the experiment results in relation to expected results from theoretical physical grounds, as well as outlining a few sensitivity experiments done on key parameters. The paper is generally well written and understandable. The science is reasonably interesting and good enough to warrant publication in The Cryosphere. I must say though that the paper quite esoteric, caters to a very narrow audience, and has relatively weak conclusions. As with all idealised, large-scale sea-ice experiments, this one suffers from the fact that comparison to theory, as well as the generalisation of the results, is very difficult.

It is interesting to see a new implementation of the MEB rheology, which as far as I know has so far only been implemented by Danserau et al (2016) and Rampal et al (2019). Also, even though the setup is virtually the same as that of Dansereau et al (2017), the authors of this paper still to point out some interesting characteristics of the MEB rheology, as their approach is sufficiently different from that of Dansereau et al (2017). The main weakness of the paper is that even though there are some interesting points made (e.g. about the lack of strain hardening and the presence of numerical errors), then those are largely lost to other less interesting aspects (e.g.
attempts to estimate physical parameters which should be estimated from a realistic setup). Ideally, the authors should reassess what is really interesting and novel here and focus on those aspects.

We thank the referee for his or her thorough review of the manuscript and constructive comments.

Other general comments:

\*) The abstract should be rewritten, as it does not fit well enough the contents of the paper itself.

>> The abstract was re-written to better reflect the manuscript's content, also as per the comment of reviewer #1.

\*) The description of the MEB model is much too detailed. It should suffice to briefly describe those parts of the model that are particularly relevant to the experiments conducted here, as well as those points where the current implementation differs from that of Dansereau et al and Rampal et al. The differences should also be justified.

>> We have simplified somewhat the description of the model by eliminating some of the repetitions. The new model description is still more detailed than what the reviewer would like to see however. Sea ice models have been developed mainly by engineers (e.g. Hibler, Flato, Weiss, Sulski); yet they are used by a climate community composed mainly of physicists. While we agree that the model description could be shortened and make reference to previous work, we decided to present a detailed (stand-alone) description of the model, including details that are often trivial to engineers but less so for the climate community. This point is evident when looking at the development of sea ice modeling as used by the climate community in the last 40 years: as of today, most Global Climate Models use a modification of the standard VP model of Hibler published in 1979. Our goal is to make the model physics more accessible to the broader community such that improvement in future GCM relates to model physics (e.g. using the Elastic Anisotropic Plastic (EAP) rheology, the MEB rheology, VP rheology with Mohr-Coulomb and dilatation or the Elastic Cohesive rheology, etc.), not just the numerics.

\*) The discussion of the cohesion (3.2.1) should take the following into account: Cohesion scales with the model resolution, so you cannot recommend one cohesion value for all resolutions (Weiss et al., 2007, Schulson et al 2009, Rampal et al 2016)

>> We agree that the cohesion scales with the scale of features as documented in Weiss et al., (2007) and Schulson et al., (2009). Here, we propose a cohesion value that is consistent with ice bridge observations, which are at a scale typical of current sea ice models (10-100km). While Rampal et al., (2016) document the scaling of deformations in the MEB model, it is not clear that the model resolution impacts the cohesion of sea ice. This was tested by repeating the simulation using different spatial resolution, showing no change in the results.

Comparing ice bridges across different straits should take the ice thickness into account. Ice bridges longer than 100 km were probably a regular feature of the Kara Sea fast-ice cover (Divine et al., 2005, Olason, 2016) - although this is changing with a thinning ice cover there.

>> We agree with the reviewer. Note that we now use vertically integrated strength parameters. These changes in ice thickness will influence the ice bridge stability in the model, as it should be.

\*) Angle of internal friction (3.2.2): I'm not convinced this is an appropriate setup to discuss the internal angle of friction. I would at least have wanted to see variations in the domain geometry, or better yet a model run with the setup from Ringeisen et al (2019).

>> We agree. Our findings described here raise interesting questions that we are currently working on, using the same numerical set-up as Ringeisen et al, (2019), and will be publish in a subsequent paper. This is clarified in the revised manuscript at L487-489:

*"This raises the question whether the lines of fracture may be influenced by the stress correction path*

*used in the damage parameterization, which determines the stress state memory. These questions will be addressed using a uniaxial compression setup (such as in Ringeisen et al., 2019) in future work."*

\*) Conclusions: You have a tendency to restate speculations from the text as demonstrable conclusions in the conclusions section. This is a serious fault which cannot be allowed to stand.

>> The conclusions are now clearly differentiated between demonstrated results, speculated results and future work.

Specific comments:

L45: "minimum viscosity" should be "maximum viscosity"

>> This line was removed in the revised manuscript.

L72: The term "brittle" refers to a certain type of plasticity, so you cannot contrast brittle and plastic (as in "i.e. Brittle [sic] in the MEB, plastic in the EVP"). Sea ice is a brittle plastic, but it can be argued that the (E)VP gives a (too) ductile behaviour to accurately represent sea ice.

>> Brittle is not a type of plasticity, but refers to a mode of fracture with little prior plastic deformation (see Crandall et al., 1978 for a reference book) before fracture. However, we agree that "brittle" should not be used in contrast to plasticity, as brittle materials can undergo plastic deformation after fracture (e.g. glass). This is corrected in the revised manuscript at L51-52:

*"The simulated stable ice arches in Dansereau et al. (2017) are located downstream of either Smith Sound or Kennedy channel (see orange curve in Fig 1). These locations differ from the observed ice arch positions in Nares Strait upstream of these channels (e.g., see Fig 1) or in the Lincoln Sea (Vincent, 2019), which are well reproduced by standard VP or EVP models (e.g., Dumont et al., 2008; Rasmussen et al., 2010)."*

L75: It should be "an MEB rheology", not "a MEB rheology"

>> Corrected as suggested by the reviewer.

L75: "implemented in an Eulerian finite difference VP model" - you should elaborate to make this clearer. I didn't understand what you meant before reading your section 2.3.

>> Clarified as suggested by the reviewer. The revised text now reads:
*"we present our implementation of the MEB rheology on the FD numerical framework of the McGill VP sea ice model."*

L122: Strike ", or creep," as creep usually refers to very slow viscous deformation of the ice (ductile deformation), but the viscous part of the MEB represents the stress
relaxation that occurs after a brittle rupture.
>> The viscous term in the MEB model is always present with different relative magnitude, not only post-fracture. While the viscous deformation is very small before the fracture, it is present and slowly dissipates the elastic stress memory, stabilizing the model. This is clarified at L119-124 and L497-498 in the revised manuscript. For example, using the model without the damage parameterization, a sustained internal stress of 50kN/m induces a viscous creep deformation of order $10^{-5}$.

L125: Rewrite the sentence "This brittle component : : :" emphasizing that both models are plastic, but MEB is brittle while VP is ductile.

>> This was rephrased, as suggested by the reviewer. Note however that a brittle material is defined as a material that breaks with little prior elastic deformation and without significant plastic deformation. In the MEB model, the development of the fracture is not instantaneous and the damage increases over several time steps during which the deformations progress but not the stress state. As such, as in the VP model, the development of brittle fractures in the MEB model is parameterized as a plastic deformation. The models differ in the deformation rule (a flow rule is used in the VP model, while the stress-strain relation remains visco-elastic in the MEB) and in the post-fracture deformations. We add these clarifications at L203-218 in the revised manuscript.

Section 2.2.1 seems unnecessary (or at least needlessly long) as it's a repetition of previous work. Ditto for section 2.2.2, except for the point about the lack of strain hardening, a point I don't recall being discussed before in the literature. The authors would do well to develop this point further and highlight it in their experiments.
>> We removed self-repetition in this section but did keep some material included in earlier work for the sake of completeness and for the general reader. We also added clarifications on the strain hardening statement, that relates to the use of vertically integrated equations rather than to an actual hardening of the ice material, which is not parameterized in the model.

L207: Replace lowercase nphi with uppercase nPhi (as well as throughout the rest of the text I believe)

>> Corrected as suggested by the reviewer.

L223: Missing unit for nlambda_0

>> The units (s) were added, as suggested by the reviewer.

L224: I don't think it's true that for high enough nlambda_0 MEB becomes EB. At any rate, Bouillon and Rampal (2015) and Rampal et al., (2016) are the wrong references for such a statement.

>> The reference has been corrected, as suggested by the reviewer. This can be seen in a simple scale analysis. In the limit where $\lambda_0$ tends to infinity, the viscous relaxation term tends to zero, which makes the system of equations reduce to that of the EB model. This is, for example, mostly the case in landfast ice, where $\lambda = \lambda_0 = 10^5$, making the viscous term orders of magnitude smaller than other terms. In damaged ice, however, $\lambda$ is reduced by 8-9 orders of magnitude such that the viscous term becomes important unless an unrealistically high $\lambda_0$ is used.

Section 2.3: After 5 pages of model description we (finally) have something novel. I dare say only the most attentive reader will make it this far, which would be a pity. You should highlight sections 2.3.1 and 2.3.3 and severely shorten everything else in section 2.

>> As we specified above, while we agree that the model description could be shortened and make reference to previous work, we decided to present a detailed (stand-alone) description of the model, including details that are often trivial to engineers but less so for the climate community. Our goal is to make the model physics more accessible to the broader community such that improvement in future GCM relates to model physics (e.g. EAP, MEB, VP with Mohr-Coulomb and dilatation, Elastic

Cohesive, etc.), not just the numerics.

L293: Both Rampal et al. (2016) and Dansereau et al. (2016) use the finite element method for the spatial discretisation. Rampal et al., however, use a Lagrangian advection scheme.

>> This was clarified at L220-225 of the revised manuscript, which now reads:

"… *and presents a significant change from previous implementations that use Finite Element methods with a triangular mesh (Rampal et al., 2016, Dansereau et al., 2016} and/or Lagrangian advection scheme (Rampal et al., 2016)}.*"

Section 2.3.3: How does your approach differ from the fixed point iteration used by Dansereau et al. (2016)? As always for numerics the practical implications of performance and accuracy are paramount.

>> The difference is mainly in the IMplicit-EXplicit treatment of the ice thickness, concentration and damage within the non-linear iterative solver. In Dansereau 2016, the set of equation is solved using (h,A,d) from the previous time-step. Here, we use the IMEX method for these variables, where the explicit equations for (h,A,d) are moved inside the outer loop, such that the solution correspond to a fully implicit solution. This is specified at L294-295 of the revised manuscript:

*"This numerical scheme differs from that of Dansereau et al. (2017) who solve the equations using tracers (h, A, d) from the previous time level."*

Section 3: The figures should appear in the order they are referred to in the text.

>> There was a duplicate of one figure in the manuscript that led to confusion in the automatic latex-referencing. We apologize for not noticing this before submission. We have corrected all remaining issues as proposed by the reviewer. Thanks for noticing this.

L356: "This deviation results from the absence of a flow rule in the MEB model" This is a very strong statement, but you never sufficiently show this to be the case.

>> We agree with the reviewer that this is inferred but not demonstrated in the paper. This is the subject of a future paper where we clarify this statement. This is clarified at L484-489. The comments about the flow rule are re-written to better reflect our conclusions at L481-483; the revised text now reads:

*"The fact that different angles of internal friction yield the same fracture orientation (...) indicates that the orientation is not directly associated to the yield criterion in the MEB rheology (there is no flow rule in the MEB rheology)."*

L369: The statement "[n]ote that unless : : : critical stress" is true, and a key aspect of the MEB model as fracturing increases the damage but does not influence the critical stress. Changing the critical stress would be a completely different approach. You need to justify the "in contrast to real ice features" much, much better for that statement to stand.

>> This was corrected, as suggested by the reviewer. The intuitive weakening of cracked ice is already simulated by the damage parameter, which increases the effective stress resulting from a given forcing.

While it could be argued that the damage could impact the vertically integrated cohesion, it is misleading to state that this coupling between the damage parameter and the critical stress is justified from observations.

L378: I find the use of the word "point" in relation to the figures confusing. Can you use "panel" instead?

>> Corrected as suggested by the reviewer.

L397: The sentence "A physical solution : : :" cannot be allowed to stand as it is. It implies that the approach of Rampal et al. is unphysical, without stating why this is so. It also implies that the suggested approach is physical, but the support given is meagre in terms of physics. What is more, I see no physical reason to relate the yield curve parameters to ice thickness.
>> Based on this and other reviewers' comments, we opted to use the vertically integrated yield criterion. This solves the issue discussed here and in the original manuscript, as it was mainly the consequence of using vertically integrated stress but a non-integrated yield criterion. That being said, we are not aware of a physical motivation for the inclusion of the pressure term in the momentum equation as in Rampal et al., 2016. It is also explicitly specified in their paper that this term is used to prevent excessively large ridges, therefore is included for numerical reasons, not for physical reasons.

L402: I found this discussion interesting, but it's tagged onto a very descriptive part of the paper and unlikely to receive much attention as it stands.

>> We agree with the reviewer. We have now created a new section 4 where we collated the text related to the error analysis.

L536: I would not describe sea ice as being granular here. It can be, but the central pack, which MEB should describe, is not - nor is the unbroken ice cover the fractures are propagating through.

>> Corrected as suggested by the reviewer.

L535: Here you state that the discrepancies between simulated and expected fracture angles are due to the use of a scalar damage parameter. However, in the text itself, you appropriately say that you _speculate_ that this is the case. You should also use this formulation in the conclusions, as you never conclusively show why you don't get
the fracture angles you expect.

>> We agree with the reviewer. This has been rephrased in the revised manuscript.

L544: You never showed that these errors are not detectable in a different configuration.

>> We do not claim to demonstrate it here, but explain that it is not possible to quantify it in non-symmetric simulations. The wording is modified to better reflect this in the conclusions of the revised manuscript.

L547: You don't show that the use of a damage tensor and a different stress correction scheme would solve the problem.

>> This sentence states this as a "possible solution". We removed this suggestion and leave it for future

work in the revised manuscript.

L558: Recommendations for who? You've only shown idealised experiments, so it is very hard to recommend anything to people wanting to run a realistic setup.

>> We have removed the bullet-point recommendations as they are repeating the previous text. However, these conclusions are not only meaningful in our model setup; they directly relate to the damage parameterization itself. For instance, we clearly show a mathematical instability in the damage parameterization equations, which are the basis of the MEB model. There are no reasons to believe that this instability is absent in other implementations, unless some undocumented dissipating factors are used.

L559: Again, in the idealized setup you need this - but what is the impact in other scenarios? You should at least make that distinction clear.

>> This has been rephrased to focus on the need to mitigate the instabilities rather than giving a specific tolerance criterion. As stated above, there is no reason to believe that a mathematical instability would not be present in other simulations, unless a different stress correction scheme is used.

L562: You never show this to be the case, it, therefore, doesn't belong to the conclusions, and certainly not to your recommendations.

>> As stated in earlier comments, we are now using the vertically integrated cohesion in our model. This recommendation is now removed, and the vertically integrated cohesion is now part of the parameterization and should have been present in the first place.

Crandall, S. H., N. C. Dahl and T. J. Lardner, eds., An Introduction to the Mechanics of Solids, 2nd ed., McGraw-Hill, New York, 1978.

Dansereau, V., Weiss, J., Saramito, P., Lattes, P., and Coche, E.: A Maxwell-Elasto-Brittle rheology for sea ice modeling, Mercator Ocean Quarterly Newsletter, pp. 35–40, 2015.

Dansereau, V., Weiss, J., Saramito, P., and Lattes, P.: A Maxwell elasto-brittle rheology for sea ice modelling, The Cryosphere, 10, 1339–1359, https://doi.org/10.5194/tc-10-1339-2016, 2016.

Dansereau, V.,Weiss, J., Saramito, P., Lattes, P., and Coche, E.: Ice bridges and ridges in the Maxwell-EB sea ice rheology, The Cryosphere, 11, 2033–2058, 2017.

Rampal, P., Bouillon, S., Ólason, E., and Morlighem, M.: neXtSIM: a new Lagrangian sea ice model, The Cryosphere Discussions, 9, 735 5885–5941, https://doi.org/10.5194/tcd-9-5885-2015, http://www.the-cryosphere-discuss.net/9/5885/2015/, 2015.

Rampal, P., Dansereau, V., Olason, E., Bouillon, S., Williams, T., and Samaké, A.: On the multi-fractal scaling properties of sea ice deformation, The Cryosphere Discussions, 2019, 1–45, https://doi.org/10.5194/tc-2018-290, https://www.the-cryosphere-discuss.net/tc-2018-290/, 2019.

Ringeisen, D., Losch, M., Tremblay, L. B., and Hutter, N.: Simulating intersection angles between

conjugate faults in sea ice with different viscous–plastic rheologies, The Cryosphere, 13, 1167–1186, https://doi.org/10.5194/

Schulson, E., & Duval, P. (2009). Creep and Fracture of Ice. Cambridge: Cambridge University Press. Doi: 10.1017/CBO9780511581397

Weiss, J., Schulson, E. M., and Stern, H. L.: Sea ice rheology from in-situ, satellite and laboratory observations : Fracture and friction, Earth and Planetary Science Letters, 255, 1–8, https://doi.org/10.1016/j.epsl.2006.11.033, 2007.

---

## Referee Report (RR1)

Landfast sea ice material properties derived from ice bridge simulations using the Maxwell Elasto-Brittle rheology

The manuscript describes an implementation of the MEB rheology in the Mcgill finite Difference framework. It describes an idealized test case that is close to real cases of the Canadian Arctic Archipelago. The manuscript describes how ice arches are formed and breaks down, how the damage and stress fields are formed and evolves. The article is in general well written and with a few minor changes it should be ready for publication.

The main concern of the model is artifacts in the damage fields that are referred to as the numerical issues. I don't think that this is a show stopper for the manuscript but it is an issue that needs to be addressed in order to properly utilize the model. One concern is whether this influences the results described in this manuscript.

Minor changes

Line 45 would leave out "constituting the cornerstone". I agree that the damage parameter is important but I (this is a personal opinion) don't think that it is necessary to put this in.

Line 62-66 I think that this is more a conclusion than an introduction part and it should be left out. This wrap up of conclusions are already

Line 281 I was a little confused by the (a,A,d) parenthesis. Maybe just use "With nx center points of in the x direction…

Line 334 I  assume that it is figure 7a that is referred to. Please add the correct subfigure as well as the figure number

Line 399 ice arche, typo

---

## Author Response (AR2)

Dear Editor,

We are pleased to resubmit a revised manuscript of our paper entitled "Landfast sea ice material properties derived from ice bridge simulations using the Maxwell Elasto-Brittle rheology" by Mathieu Plante, Bruno Tremblay, Martin Losch and Jean-François Lemieux.

We would like to thank the reviewers for their useful comments and suggestions. We have addressed all comments as per the reviewer's suggestion. You will find below our specific responses to each of the comments.

Regarding the comment on the plain-strain assumption used in Dansereau et al., 2016 (last comment, reviewer #2), this stems from a misunderstanding on our part about the actual equations that were used in their paper. Indeed, the authors presented the general 3D stress-strain relationships in their paper. These equations have the same form as the stress-strain relation for a 2D material using the plain-strain assumption. Given that they did not give the stress-strain relations under plane-stress assumption, we assumed (mistakenly) that they were not used in the model. The offensive comment had already been removed from the previous re-submission. We hope that this will satisfy you and the reviewer.

Thanks again for your consideration.

Sincerely,

Mathieu Plante,
on behalf of all the authors.

Note :

- The referee comments are shown in black,
- The authors answers are shown in blue.

**Answers to tc-2019-2010 RC1**

May 4[th], 2020

Landfast sea ice material properties derived from ice bridge simulations using the Maxwell ElastoBrittle rheology

The manuscript describes an implementation of the MEB rheology in the Mcgill finite Difference framework. It describes an idealized test case that is close to real cases of the Canadian Arctic Archipelago. The manuscript describes how ice arches are formed and breaks down, how the damage and stress fields are formed and evolves. The article is in general well written and with a few minor changes it should be ready for publication
.
The main concern of the model is artifacts in the damage fields that are referred to as the numerical issues. I don't think that this is a show stopper for the manuscript but it is an issue that needs to be addressed in order to properly utilize the model. One concern is whether this influences the results described in this manuscript.

>> We agree that this is important. The numerical issue described in the manuscript does influence the results for large compressive stress states. In the manuscript, we limited the discussion to results to where the numerical issue is not present. This is clarified in the discussion at L493, which reads:

" *This limits the current analysis to short-term simulations in which this issue remains negligible*. "

Note that the instability in the damage parameterization and modifications to improve it is being investigated in more details and will be submitted shortly for publication.

All minor changes listed below were corrected as suggested by the reviewer.

Minor changes

Line 45 would leave out "constituting the cornerstone". I agree that the damage parameter is important but I (this is a personal opinion) don't think that it is necessary to put this in.

Line 62-66 I think that this is more a conclusion than an introduction part and it should be left out. This wrap up of conclusions are already

Line 281 I was a little confused by the (a,A,d) parenthesis. Maybe just use "With nx center points of in the x direction…

Line 334 I assume that it is figure 7a that is referred to. Please add the correct subfigure as well as the figure number

Line 399 ice arche, typo

**Answers to tc-2019-210 RC2**

May 4[th], 2020

This review has been done jointly with V. Dansereau.

Following the comments of the different referees, the authors carefully answered to most of them, and substantially modified the original manuscript, which, in our opinion, significantly improved. We have now only one comment that should be taken into account before publication, plus minor comments and some clarifications on their answer. Once these comments taken into account, this manuscript should be suitable for publication in The Cryosphere.

>> Mechanical redistribution is taken into account in our simple 1-category model (i.e. ice or open water). When $A=1$ and sea ice convergence occurs, the mean ice thickness increase (see continuity Eq. 4 in the manuscript), but since $A=1$ is capped at one, this leads to the actual thickness of ice in a grid cell ($h/A$) to increase, i.e. ridging. A simple 1-category model does not resolve the ITD per se, unless the variability in ice thickness is resolved (i.e. unless the model is run at $O(1m)$ resolution, at which sea ice no longer behaves as a 2D material.

This mechanical redistribution scheme is still not mentioned in section 2.1 of the revised manuscript and, apparently, nowhere else (?). This should be corrected.

>> This was added in the revised manuscript at L100-101. Thanks for pointing this out. It now reads:

*"Mechanical redistribution (i.e. ridging) is taken into account by capping the ice concentration at 1 (or 100%) in convergence. As the mean ice thickness $h$ is allowed to grow, the capping increases the actual ice thickness (Schulkes et al., 1995).*

In the meantime, we do not agree with their view that a 1-category model does not resolve the ITD: there is no a priori and unique spatial scale associated to the Ice Thickness Distribution, which can either be calculated (a PDF) from a modelled field or set of observational data. The difference in the scale of measurements and that of current model resolution however do differ substantially and so one must be careful when comparing modelled and observed ITD.

Please note that we are not talking here about the subgrid-scale parametrization often referred to as ITD (the equations of which are typically cast in 2-dimensions). This is a minor comment.

>> We agree. We were indeed referring to the subscale parameterization in our comment to the reviewer. Note that the ITD is not discussed in the paper.

>> The paper of Dansereau et al. (2016) have a factor of $1/(1+v)(1-2v)$ in the stress-strain relationship,

indicating that the authors have used the plane-strain assumption. Dansereau et al. (2017) however do use the plane stress assumption, as pointed out by the reviewer. In the revised manuscript, we remove the reference to the plane-strain assumption in Dansereau et al. (2016), for conciseness.

This statement is still false:
In Dansereau et al., 2016, p. 1343, section 3.1, the "factor" indicated is nu/((1+nu)(1-2nu)).
Please note that at the top of this paragraph, you find the sentence "Here we apply this idea of stress dissipation to a TWO- or THREE-dimensional, compressible, elastic continuous solid (…). Equation 4 just below this sentence does not make any distinction between a 3D or 2D case. Then, the equation for the elastic stiffness tensor, given just below, does not make any distinction between the 2D plane stresses or the plane strains approximation. Indeed, it is written "the (dimensionless) elastic stiffness tensor K is defined in terms of nu, (…) such that for all THREE-dimensional symmetric tensor epsilon [deformation rate] (…)" and there comes the factor you are referring to, which is the indeed the right one for linear elasticity in 3D. Thus, in this section of the paper, no assumption is chosen between 3D, 2D plane stresses or 2D plane strains to represent sea ice on the large scale. This assumption is rather stated in section 4, but the factor is not explicitly written there. Again, the correct 2D, plane stress version of this factor was used in the simulations of Dansereau et al., 2015, 2016 and 2017.

>> This stems from a misunderstanding on our part about the actual equations that were used in their paper. Indeed, the authors presented the general 3D stress-strain relationships at the beginning of their paper. These equations have the same form as the stress-strain relation for a 2D material using the plain-strain assumption. Given that they did not give the stress-strain relations under plane-stress assumption, we assumed (mistakenly) that they were not used in the model. The offensive comment had already been removed from the previous re-submission. We hope that this will satisfy you and the reviewer.

[revised manuscript text omitted]